# What is Missing? Explaining Neurons Activated by Absent Concepts

**Robin Hesse** [1 2]  **Simone Schaub-Meyer** [2 3]  **Janina Hesse** [4 5 6]  **Bernt Schiele** [1]  **Stefan Roth** [2 3]

## Abstract

Explainable artificial intelligence (XAI) aims to provide human-interpretable insights into the behavior of deep neural networks (DNNs), typically by estimating a simplified causal structure of the model. In existing work, this causal structure often includes relationships where the presence of a concept is associated with a strong activation of a neuron. For example, attribution methods primarily identify input pixels that contribute most to a prediction, and feature visualization methods reveal inputs that cause high activation of a target neuron – the former implicitly assuming that the relevant information resides in the input, and the latter that neurons encode the presence of concepts. However, a largely overlooked type of causal relationship is that of *encoded absences*, where the absence of a concept increases neural activation. In this work, we show that such missing but relevant concepts are common and that mainstream XAI methods struggle to reveal them when applied in their standard form. To address this, we propose two simple extensions to attribution and feature visualization techniques that uncover encoded absences. Across experiments, we show how mainstream XAI methods can be used to reveal and explain encoded absences, how ImageNet models exploit them, and that debiasing can be improved when considering them.

## 1. Introduction

Two of the arguably most important methods in explainable artificial intelligence (XAI) for computer vision – namely,

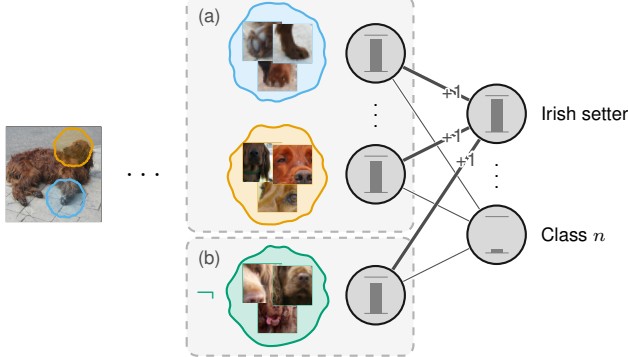

*Figure 1.* **Encoded absence in image classification.** *(a)* The model detects concepts present in the input image that are prototypical for the target class (*e.g.*, the snout and feet). *(b)* The model can additionally encode the absence of snouts from other dog species to enhance evidence for the "Irish setter" class.

attribution and feature visualization techniques – primarily associate the activation of a neuron with the *presence* of specific concepts. For instance, attribution methods highlight which features present in the input have contributed to the activation of a neuron of interest, and feature visualization methods find input patterns whose presence maximizes a neuron's activation.

However, in biological neural networks, presences are only one side of the story. Equally important are *absences*, which often serve as powerful reasoning cues. *E.g.*, in clinical diagnosis, humans may pay closer attention to the absence of specific symptoms than to the proper functioning of dozens of physiological processes. Likewise, the Hassenstein–Reichardt model (Egelhaaf et al., 1989) describes neurons in the *Drosophila melanogaster* that are activated by the presence of rightward motion in combination with the absence of leftward motion, enabling the fly to distinguish rightward motion from predators whose looming movement produces motion in multiple directions (*cf.* Appendix A.1).

While there are isolated indications that deep neural networks (DNNs) also exploit information conveyed by the absence of concepts – appearing, *e.g.*, in logical explanations or in analyses of individual circuits – these observations remain fragmented and lack a standardized notion of encoded absence (*cf.* Section 3). To our knowledge, there is no systematic study of how absent concepts are encoded, explained, or exploited in modern DNNs. As a result, an

---

[1]Max Planck Institute for Informatics, SIC [2]Department of Computer Science, Technical University of Darmstadt [3]hessian.AI [4]Leibniz Institute for Resilience Research [5]Institute for Quantitative and Computational Biosciences, Johannes Gutenberg University Mainz [6]University Medical Center Mainz. Code at: github.com/visinf/what-is-missing. Correspondence to: Robin Hesse <rhesse@mpi-inf.mpg.de>.

*Proceedings of the 43rd International Conference on Machine Learning*, Seoul, South Korea. PMLR 306, 2026. Copyright 2026 by the author(s).

important aspect of model behavior remains largely unexplored, with potential implications for robustness and bias.

In this work, we close this gap by showing how standard attribution and feature visualization approaches can be used to illuminate encoded absences in DNNs, *i.e.*, concepts *not* visible in the input but still causally linked to the prediction (exemplified with image classification models). By doing so, we show that absences are especially relevant for fine-grained classification, where subtle differences matter: distinguishing an Irish Setter from a Sussex Spaniel benefits not only from detecting Setter-specific features but also from confirming the *absence* of Spaniel-specific ones (see Figure 1). Further, we show how our proposed modifications can be used for debiasing models based on absences. More specifically: *(i)* We formally define encoded absences as a largely overlooked causal relationship in DNNs. *(ii)* We illustrate how DNNs encode such absences at a mechanistic level. *(iii)* We analyze why mainstream explanation methods fail to capture encoded absences in their standard form, and show how attribution and feature visualization can be adapted to reveal them. *(iv)* We empirically validate our findings, demonstrating how absences are used in image classification and how they can be leveraged for debiasing.

## 2. Encoded Absences

Before discussing related work, we introduce a causal formulation of encoded absences and outline how they can arise in neural representations. This grounds the notion of encoded absence in a causal perspective, clarifies why it constitutes a distinct and relevant explanatory relationship, and provides the conceptual foundation for the methods and analyses introduced later.

### 2.1. A causal perspective on encoded absences

The goal of XAI can be reframed as finding a *simplified* approximation of the DNN's underlying causal structure (Hesse et al., 2023; Carloni et al., 2025). While the *true* causal structure is embodied by the DNN itself, its complexity typically exceeds human understanding. Thus, a "simplified" structure refers to one that enables a human to understand the model sufficiently to answer task-specific questions of interest. Since the appropriate level of simplification depends on both the user and the task (Tomsett et al., 2019), a wide range of causal abstractions could be relevant – and should be explored within XAI research. Formally, a feed-forward DNN $f : \mathbb{R}^n \mapsto \mathbb{R}$ can be expressed as a structural causal model (SCM) $\mathfrak{C} := (\mathbf{S})$ (Peters et al., 2017) with structural assignments $\mathbf{S}$ defining each intermediate representation as a deterministic function of its parents, *i.e.*, $z^{(1)} := f^{(1)}(x)$, $z^{(2)} := f^{(2)}(z^{(1)})$, $\ldots$, $y := f^{(n)}(z^{(n-1)})$, where $x$ is the input; the noise variables usually found in SCMs are set to zero for simplicity. In XAI,

we seek a simplified SCM $\mathfrak{C}'$ that approximates the original SCM $\mathfrak{C}$ in a way that preserves task-relevant causal relationships while improving human interpretability (Hesse et al., 2023; Carloni et al., 2025). *E.g.*, in the case of a simple gradient-based attribution method (Simonyan et al., 2014), $\mathfrak{C}'$ would be a linear approximation of the structural assignment $y := f(x)$, where each feature $x_i$ is associated with a causal influence estimated by $\frac{\partial f(x)}{\partial x_i}$ (see Appendix A.3 for feature visualization and counterfactual explanations).

A less studied causal relationship in XAI involves concepts whose *absence* causes higher activations, or vice versa, whose presence *suppresses* the activation of a specific internal neuron $z_j$ or output $y$. Formally, let $C_{\hat{x}} \in \{0, 1\}$ denote a binary variable indicating whether the concept $\hat{x}$ is present in an input $[x, C_{\hat{x}}]$; here, a concept is a family of input patterns that share a common effect on a neuron (*e.g.*, images of "dog snouts", *cf.* Appendix A.2). An inhibitory relationship holds for neuron $z_j$ in layer $l$ whenever $f_j^{(l)}\big(do(x := [x, C_{\hat{x}} = 1])\big) < f_j^{(l)}\big(do(x := [x, C_{\hat{x}} = 0])\big)$, *i.e.*, introducing the concept in the input $x$ via $do(x := [x, C_{\hat{x}} = 1])$ decreases the activation. Intuitively, such interventions reveal patterns that actively suppress a neuron's activation, akin to the example of the Hassenstein–Reichardt detector, where the opposite motion direction inhibits the response. Section 4 outlines how to uncover this causal relationship.

**Definition 2.1** (Encoded Absence). If there exists a concept $\hat{x}$ whose presence causes the activation of a neuron $z_j$ in layer $l$ to decrease, *i.e.*, $f_j^{(l)}([x, C_{\hat{x}} = 1]) < f_j^{(l)}([x, C_{\hat{x}} = 0])$, we say that the neuron $z_j$ encodes the *absence* of said concept $\hat{x}$ in the input context of $x$.

### 2.2. A mechanistic perspective on encoded absences

Having established that encoded absences can contribute to more complete explanations, we now present a constructive existence proof demonstrating that neural networks are capable of implementing such neurons.[1]

**Proposition 2.2.** *DNNs can implement neurons $z_j$ that encode the absence of a concept $\hat{x}$ in the input context of $x$.*

For our construction, we assume that in layer $l - 1$ each neuron encodes the *presence* of one or multiple concepts $\{\hat{x}, ...\}$; if a neuron in $l - 1$ already encoded the absence of a concept, the proof would be trivially complete. A simple way to construct a neuron $z_j$ in layer $l$ that encodes the absence of concept $\hat{x}$ encoded in layer $l - 1$ involves two components: *(i)* negative weights connecting neurons in $l - 1$ that encode the *presence* of $\hat{x}$ to $z_j$, and *(ii)* a source of positive potential to ensure that $z_j$ is activated when $\hat{x}$

---

[1]While previous work has already shown that neural networks are capable of implementing logical NOT operations (Dukor, 2018) – which is similar to encoding absences – we offer a more rigorous proof here.

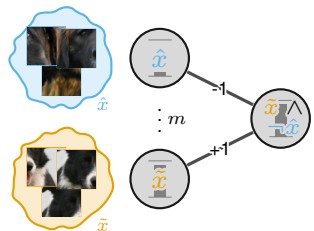

*Figure 2.* **A mechanistic process to encode the absence of a concept.** A neuron encoding the *absence* of concept $\hat{x}$ (*i.e.* $\neg\hat{x}$) can be implemented by having a negative connection to a neuron encoding $\hat{x}$ and a positive potential through, *e.g.*, another activating concept $\tilde{x}$ (*i.e.*, the output encodes $\tilde{x} \wedge \neg\hat{x}$).

is *absent*. When both conditions are met, $z_j$ will produce a higher activation if $\hat{x}$ is *absent*, and a lower activation if $\hat{x}$ is *present* – effectively encoding the *absence* of $\hat{x}$. This construction is illustrated in Figure 2. The positive potential can be supplied, *e.g.*, by using the activation of another concept $\tilde{x}$ in $l-1$. As a result, $z_j$ jointly encodes the *presence* of $\tilde{x}$ and the *absence* of $\hat{x}$. In Appendix A.4, we provide additional mechanisms to encode absences and extend the idea to polysemantic neurons, respectively, concepts that lie on arbitrary feature space directions (Elhage et al., 2022).

## 3. Related Work

**Explaining encoded absences.** Ideas related to DNNs exploiting information conveyed by the absence of concepts have appeared in prior work. However, these efforts remain fragmented, rely on differing notions of what constitutes an absence, or are limited to inherently explainable models.

Explanations based on logical compositions have shown that neurons can be associated with logical forms that include a NOT operation. Mu & Andreas (2020) build on Network Dissection (Bau et al., 2017) by thresholding activations into binary masks and searching (via beam search) for compositional logical expressions over concept masks that maximize the IoU with the neuron's activation mask. Rosa et al. (2023) extend this line of work by accounting for different activation ranges. However, in these approaches, NOT reflects (spatial) non-overlap with a concept mask in the probing dataset, and does not by itself establish that the presence of the concept causally suppresses activation. This contrasts with our notion of encoded absences, which requires evidence of active inhibition (see Section 5.3). Olah et al. (2020) identify specific circuits in which negative connections give rise to inhibitory signals, closely aligning with our mechanistic perspective (Section 2.2). However, these findings are anecdotal and limited to small-scale circuits. Dhurandhar et al. (2018) propose contrastive explanations based on missing information; however, for image data their analysis is restricted to binary digit datasets, where absence is equated with black pixels. In natural images, black pixels can themselves carry semantic meaning, making it unclear

whether the model relies on the presence of black pixels or the absence of a concept. Also loosely related is the notion of *criticism* (Kim et al., 2016), where explanations include samples that are not well captured by class-specific prototypes. While this offers insight into corner cases missed by prototype explanations, the method does not aim to identify *inhibitory* signals, which is the focus of our notion of encoded absence. Oikarinen & Weng (2024) consider the full activation range for explanations; however, for ReLU-based models they operate on post-ReLU activations, which collapse inhibitory signals in the negative range to zero, making them indistinguishable from simply non-activating concepts. Beyond post-hoc explanations, there are inherently interpretable architectures that explicitly incorporate negative reasoning. For example, Prabhushankar & AlRegib (2021) and Singh & Yow (2021) design models that make predictions by leveraging absences. As these approaches require specialized architectures and cannot be applied post hoc to arbitrary networks, we consider them complementary but beyond the scope of this work.

**Limitations of mainstream XAI methods.** While related ideas exist in more specialized explanation settings, encoded absences remain largely unaddressed by mainstream XAI methods for image classification, such as the below.

*Attribution methods* estimate how important each input feature is to a (possibly intermediate) model output (Bach et al., 2015; Simonyan et al., 2014; Sundararajan et al., 2017). By construction, they highlight only features present in the input. As a result, explaining that the absence of a concept contributed to a prediction cannot be achieved directly. While negative attributions (*e.g.*, Lundberg & Lee, 2017) can sometimes be interpreted as inhibitory signals, many methods focus solely on attribution magnitude (Simonyan et al., 2014; Srinivas & Fleuret, 2019; Yang et al., 2023), discarding the sign and thus obscuring inhibitory effects.

*Feature visualization* aims to reveal concepts encoded in individual neurons by finding inputs that strongly activate them (Erhan et al., 2009; Olah et al., 2017). For neurons encoding absences, however, maximizing activation yields visualizations that explicitly exclude the suppressing concept rather than depicting it. Consequently, standard feature visualization through maximization cannot faithfully explain neurons whose function relies on concept absence.

*Concept discovery methods* extend feature visualization to more sophisticated concepts, *e.g.*, by using feature space directions or textual descriptions (Fel et al., 2023; Oikarinen & Weng, 2023; Ahn et al., 2024; Kim et al., 2018). Similar to feature visualization, most concept discovery pipelines are designed to identify concepts associated with high activation or positive relevance. As a result, they are naturally suited for explaining encoded presences, whereas explaining encoded absences requires additional adjustments analogous

to our adjustments for feature visualization in Section 4.

*Counterfactual explanations* identify why one prediction was made instead of another by contrasting specific samples or classes (Goyal et al., 2019; Wang et al., 2023; Guidotti, 2024; Verma et al., 2024), *e.g.*, by swapping content. While our work is motivated by a counterfactual argument (*cf.* Section 2.1), it differs fundamentally: we contrast neuron activations against the data distribution rather than between individual samples or classes, we operate at the neuron level, and do not require "minimal" interventions. Further, the swapping of information often done in counterfactual explanations implicitly deletes content in the image under inspection, making it hard to distinguish whether a prediction relies on the *presence* of a class-specific concept, the *absence* of a competing one, or *both* (cf. Figure 1). Therefore counterfactual explanations generally cannot explain neurons encoding absences.

## 4. Explaining Encoded Absences

Equipped with the notion of *encoded absences* and an understanding of the limitations of existing work, we now show how attribution and feature visualization methods can explain encoded absences following Definition 2.1.

**Non-target attribution methods.** As discussed in Section 3, attribution methods typically compute a targeted attribution $\mathcal{A}(x, t, f)$ for an input $x$ and its target $t$ (usually the prediction $t = f(x)$ or the ground truth). Such methods highlight features present in $x$ that either excite or inhibit the neuron of interest; in principle, they can therefore capture inhibitory signals, respectively, encoded absences. However, each input image contains only a subset of all concepts relevant to a prediction. As a result, standard target attributions can reveal inhibitory effects only for concepts that are present in the input. For concepts whose absence is informative for predicting class $t$, this poses a fundamental limitation: such concepts are typically not present in images of class $t$ and therefore cannot appear in target attributions computed on those images. While this limitation is often unproblematic for presence-based reasoning – since images of the same class usually contain the relevant present concepts – it prevents standard target attribution methods from revealing most of the absence-based evidence. To address this, we compute not only the attribution for $x$, but also the attribution $\mathcal{A}(x^{(c \neq t)}, t, f)$ for class $t$ using inputs $x^{(c \neq t)}$ from other classes (or, more generally, from a diverse set of samples). Computing attributions for $t$ across such inputs ensures that all concepts influencing the prediction of $t$ are considered, including those whose absence is informative. In particular, if a model relies on the absence of a concept to predict class $t$, there will be cases where the attribution of $t$ is computed for an input in which that concept is present. According to Definition 2.1, the presence of this concept

has an inhibitory effect on the output for $t$, and the corresponding attribution will therefore be negative. We refer to this approach as *non-target attribution*, to distinguish it from the conventional target attribution.

The concrete computation of non-target attributions depends, just as in standard attribution methods, on the task; please refer to Section 5 for some examples. Moreover, since attribution methods can produce noisy results, *e.g.*, due to gradient shattering (Balduzzi et al., 2017), it is important to note that negative non-target attributions do not, by themselves, guarantee an encoded absence. In our experiments, however, we observed that this limitation did not meaningfully affect the applications we studied, a finding we further substantiate through a controlled analysis in Appendix B.5.

**Feature visualization through minimization.** As discussed in Section 3, feature visualization through maximization cannot visualize concepts whose absence is encoded by a neuron, as the inputs that maximally activate the neuron contain minimal amounts of concepts that inhibit activation. To account for this problem, we propose to use *feature visualization through minimization* to find the input $\hat{x}$ that minimizes the activation of a neuron $z_j$ (before the activation function), *i.e.*, $\hat{x} = \arg\min_x z_j(x)$. Intuitively, inputs that lead to strong negative activation highlight patterns that *inhibit* the neuron, revealing the subset of concepts whose *absence* the neuron encodes most strongly.

**Overlap with related work.** Interestingly, from an algorithmic standpoint, our proposed modifications are not entirely unprecedented. For instance, in a targeted FGSM adversarial attack (Goodfellow et al., 2015), the input gradient for a sample is computed with respect to a target class different from the true or predicted class. This can be interpreted as computing a non-target attribution, as outlined above. Similarly, Walter et al. (2026) compute attribution maps for multiple classes on the same input to obtain more class-specific explanations. For feature visualization, Olah et al. (2017) also experimented with inputs that minimize the activation of a target neuron to reveal concepts that activate a neuron to varying degrees. While it is in principle well-known that activations can be maximized or minimized, prior work treated minimization merely as a technical variant; its necessity and semantic role in encoding absences have remained largely unexplored.

That said, while these methods share the same underlying algorithms, their intent and interpretive framing differ fundamentally. To the best of our knowledge, no prior work has linked these modifications to the human-understandable notion of encoding absences, *i.e.*, concepts that are not present in the input but still causally affect the model's prediction; this omission is further reflected in our experiments, which uncover a new form of bias based on encoded absences. Our contribution lies in formalizing this perspective and

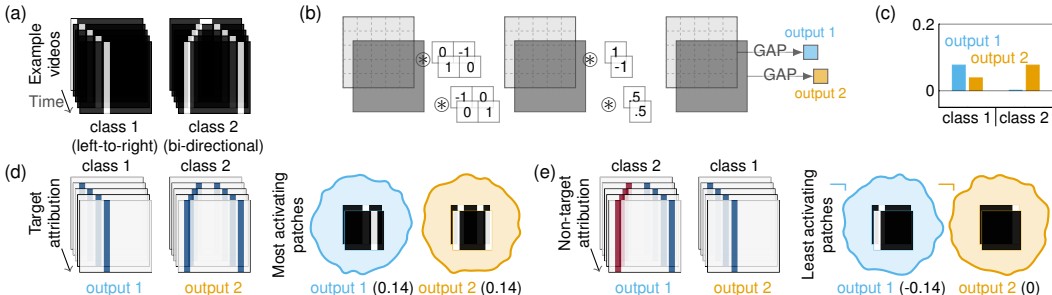

*Figure 3.* **Hassenstein–Reichardt detector experiment.** *(a)* Two example sequences showing a left-to-right and bi-directional movement. *(b)* A hand-crafted CNN to distinguish left-to-right motion from bi-directional motion. The first layer implements the spatio-temporal comparison of neighboring pixels, the second layer compares motion in opposing directions, followed by global average pooling (GAP). The first output node implements a Hassenstein-Reichardt detector (weights: 1/-1) and the second output averages both directions (weights: 0.5/0.5). *(c)* The outputs of the model for the two example sequences. *(d)* Visualizations of established XAI methods – target attribution and feature visualization via the highest activating patches, each consisting of two consecutive frames as CNN input (numbers indicate the activation strength). Both methods fail to highlight the absence encoded in the first output and thus lack a complete explanation of the CNN mechanisms. *(e)* Our proposed non-target attributions and feature visualization through minimization highlight that the first output encodes the absence of right-to-left motion.

highlighting that a complete explanation requires examining both encoded presences and absences. Importantly, our modifications are not meant to replace, but to complement, established attribution and feature visualization.

# 5. Experiments

We now empirically establish that DNNs can and do encode absent concepts, that common XAI methods struggle with them, and that our proposed modifications can illuminate these absences. We further briefly demonstrate how ImageNet-trained models make use of absences and how to debias DNNs relying on absent concepts. Since our contribution is of a conceptual nature, highlighting the relevance of absences for DNNs and XAI, we use simple experimental setups to isolate this phenomenon and leave more complex tasks for future work.

## 5.1. Hassenstein-Reichardt detector

We first revisit our example from Section 1 – the Hassenstein–Reichardt detector. As input, we generate two video sequences with left-to-right or bi-directional motion (Figure 3 (a)) and design a small hand-crafted convolutional neural network (CNN) to distinguish them (Figure 3 (b)). Using two consecutive frames as input, the first convolutional layer extracts directional motion features via spatio-temporal comparisons (followed by ReLU activation), similar to the mirror-symmetric subcircuits of the biological Hassenstein–Reichardt detector. The second layer then combines these features in two ways: one output implements a Hassenstein–Reichardt detector by subtracting motion in opposing directions (kernel of size $(C = 2) \times (H = 1) \times (W = 1)$ and weights of 1 and $-1$), activating for left-to-right motion only when right-to-left motion is absent, while the other output averages both directions and responds to bi-directional

motion (equal weights of 0.5 each). As shown in Figure 3 (c), the two outputs reliably distinguish the two sequences.

**Limitations of existing explanation methods.** In Figure 3 (d), we apply standard XAI methods – target attribution (Integrated Gradients; Sundararajan et al., 2017) and feature visualization via the highest activating patch. For the second output, which encodes the presence of both motion directions, both methods provide faithful explanations. However, for the first output, which encodes the presence of one direction and the absence of the other, they highlight only the left-to-right motion (the positive potential) and fail to reveal the encoded absence.

**Explaining absent features.** In Figure 3 (e), our proposed modifications reveal the missing inhibitory signal. Non-target attribution for the first output highlights right-to-left motion with negative attribution (red), and the *least* activating patch shows right-to-left motion as the least activating pattern. Together, these results show that the first output encodes the absence of right-to-left motion. For the second output, the attribution for the left-to-right sequence highlights the movement as expected. For the minimally activating patch, we have an activation of zero, and thus, no inhibition is happening and no absence from the dataset is encoded. To conclude, in order to obtain a complete explanation, existing and our modified XAI methods have to be used *in combination*, even for this simple model.

## 5.2. Trained toy model

We continue with a toy example in which a model classifies images based on whether they contain a green pixel (class 1) or not (class 2); see Figure 4 (a). To ensure that only the presence or absence of a green pixel is class-discriminative, the number of non-green pixels is sampled uniformly between 8 and 12 for both classes. We use a simple two-layer CNN

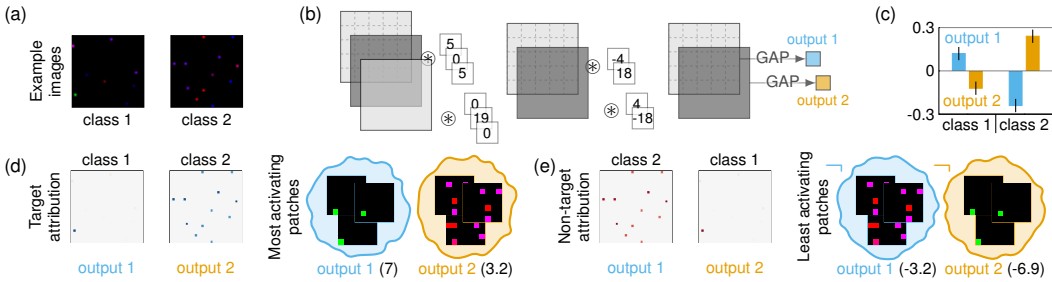

Figure 4. **Toy experiment.** *(a)* Example RGB images from class 1 (green pixel) and class 2 (no green pixel) – zoom in for better visibility. *(b)* Architecture of the used toy model with the trained weights. *(c)* Average logit output for the two output nodes for images from class 1 and 2. Confidence intervals represent two times the standard deviation. *(d)* Integrated Gradients (Sundararajan et al., 2017) target attributions for the above example images, and maximally activating patches for the two output nodes. *(e) Non-target* attributions for the two respective examples – note how the attributions switch from positive (blue) to negative (red) – and *minimally* activating patches for the two output nodes (numbers indicate the activation strength).

with $1 \times 1$ convolutions, ReLU activations, and global average pooling (Figure 4 (b); training details in Appendix B.2).

As shown in Figure 4 (c), the second output node exhibits positive activation when no green pixel is present and negative activation otherwise. According to Definition 2.1, this output therefore encodes the *absence* of a green pixel. Inspecting the learned weights in Figure 4 (b) reveals that this behavior arises from a positive connection to the channel reacting to red/blue (serving as positive potential) and a negative connection to the channel reacting to green, directly instantiating the mechanistic construction from Section 2.2. This demonstrates that even a simple DNN can learn to encode the absence of a concept.

**Limitations of existing explanation methods.** Applying mainstream XAI methods (Integrated Gradients and feature visualization) in Figure 4 (d) confirms the findings from Section 5.1: while the first output encoding the presence of a green pixel is explained faithfully, the second output encoding its absence highlights only non-green pixels providing the positive potential, but fails to reveal the causal role of the green pixel. In Appendix B.2, we further apply a counterfactual visual explanation (Goyal et al., 2019) to this setup, showing that while it identifies the green pixel as the most class-discriminative feature, it does not distinguish whether its presence or absence is relevant, making it unsuitable for explaining encoded absences.

**Explaining absent features.** We conclude our experiment by applying our *non-target* attribution and feature visualization through *minimization* in Figure 4 (e). For the second output node, now the green pixel is highlighted, faithfully explaining that the node encodes its absence. Interestingly, for the first output node, we observe inhibitory signals from the non-green pixels, indicating that the node has learned to encode the absence of red pixels although they do not contain class-discriminative information – this is further confirmed by looking at the weights of the second layer, which are just mirrored between the two nodes (with dif-

Table 1. **Quantitative evaluation of encoded absences.** We measure the activation of the 100 highest-activating images when inserting none, random, encoded logical NOTs (Mu & Andreas, 2020), most activating (*e.g.*, as used in (Ahn et al., 2024)), or least activating $48 \times 48$ patches in a random corner.

| Method | VGG19 | ResNet-50 | ViT-B/16 |
|---|---|---|---|
| None | 2.98 | 0.18 | -0.14 |
| + Random | 2.68 | 0.16 | -0.18 |
| + Logical NOT (Mu & Andreas, 2020) | 2.53→2.51 | 0.15→0.15 | – |
| + Most activating (*e.g.*, (Ahn et al., 2024)) | 3.88 | 0.25 | -0.17 |
| + Least activating *(ours)* | 0.94 | 0.03 | -0.24 |

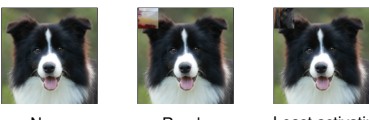

None     + Random     + Least activating

Figure 5. **Examples for Table 1.** Zoom in to see the patches.

ferent suppression strengths). Together with the previous experiment, this toy example shows that DNNs can encode absent concepts, that mainstream XAI methods struggle to expose them, and that our proposed modifications succeed.

### 5.3. Image-classification models

We now turn to a more realistic setting using ImageNet-1k (Russakovsky et al., 2015) models to test for encoded absences. According to Definition 2.1, a channel or neuron encodes the absence of a concept if the presence of the corresponding input patterns decreases its activation. To quantify this effect, we measure the drop in activation when inserting patches containing such patterns. Specifically, we compare patches obtained via feature visualization through minimization (least activating) with those derived from the only post-hoc method we are aware of with a related motivation, encoded logical NOTs (Mu & Andreas, 2020). We further report activations obtained by inserting the most highly activating patches, as done for example in (Ahn et al., 2024), to empirically demonstrate that concept discovery methods focusing on high-activation regimes are generally not suited to explaining encoded absences. To ensure that observed

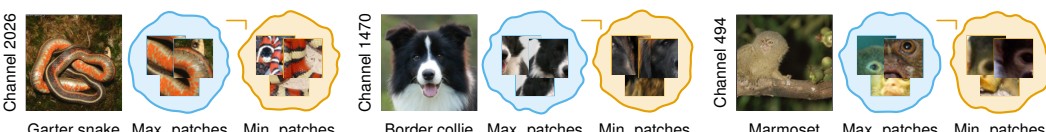

*Figure 6.* **Encoded presences (positive potential) and absences for three channels that have been found to be important for the corresponding class.** We identify channels that are important for specific classes and visualize the positive potential from this class by showing the most activating patches. Further, we show the encoded absences by showing the least activating patches. In particular, for *fine-grained* classification, encoded absences of patterns from related species seem to be used.

activation drops are not merely due to out-of-distribution artifacts, we additionally evaluate the insertion of random patches that exhibit comparable boundary discontinuities as the other methods. For each channel/neuron in the final backbone layer, we compute the average activation over the 100 most activating images and evaluate the mean drop in activation after inserting $48 \times 48$ patches, either random, encoded logical NOT, most activating, or least activating, into a random corner of each image; see Figure 5 (we report results for alternative hyperparameters in Appendix B.3). Since Mu & Andreas (2020) do not identify a logical NOT for every channel, we evaluate their method only on the subset of channels for which such a concept is found. For a fair comparison, for each of these channels, we again report the mean activations after inserting a random patch as well as after inserting a patch containing the identified logical-NOT concept (random patch → logical-NOT patch in Table 1).

Table 1 shows that random patches have little effect on activations, despite rendering the input slightly out-of-distribution, whereas the least activating patches cause strong suppression, demonstrating their inhibitory role and the encoded absence – an effect that many channels/neurons exhibit (see Appendix B.3). This suggests that encoded absences are indeed utilized in ImageNet models. Logical NOTs identified by Mu & Andreas (2020) yield no inhibitory effect beyond random patch insertions, showing that they capture a different notion of logical NOT (*cf.* Section 3). Similarly, inserting the most highly activating patches does not decrease the activation beyond the activation of random patches, highlighting that many concept discovery methods focusing on these high-activation regimes are primarily designed to explain encoded presences rather than absences.

**How absences are used.** We now seek to better understand *how* these inhibitory signals are used. While a full mechanistic understanding remains an open challenge and is beyond the scope of this paper, we provide an initial glimpse into the role of absences. To this end, for each class, we identify channels in the penultimate layer (other layers could be used as well) of a ResNet-50 (He et al., 2016) that are particularly important, similar to Hesse et al. (2025) (see Appendix B.3). Next, for each identified channel, we visualize: *(i)* the maximally activating patches from the classes the channel is

important for (to show the encoded presences, *i.e.*, the positive potential), and *(ii)* the minimally activating patches (to show the encoded absences). Across channels, we observe a recurring pattern: channels that encode the presence of concepts from the class for which the channels are important often simultaneously encode the absence of concepts from closely related classes. Three illustrative cases of this pattern are shown in Figure 6. In these examples, encoded absences appear especially useful for fine-grained classification (*e.g.*, Border Collie vs. Leonberger), where the absence of concepts associated with nearby classes provides a strong discriminative signal. This discriminative role can also improve robustness: a Border Collie with a partially occluded snout is more confidently recognized when no Leonberger-specific concepts are seen (*cf.* Figure 6, middle). While this analysis is qualitative, the consistent emergence of this pattern across multiple channels is unlikely to arise by chance, given the large number of ImageNet classes and the combinatorial space of possible class pairs. This observation also aligns, to some extent, with human intuition: distinguishing between similar categories involves ruling out nearby alternatives, whereas coarse distinctions (*e.g.*, dog *vs.* car) require such comparisons to a lesser extent.

### 5.4. Debiasing models based on encoded absences

DNNs are prone to learning spurious correlations in the training data. For instance, in the ISIC dataset (Rotemberg et al., 2021) of skin lesion images, benign samples often co-occur with colorful patches (*cf.* Figure 7 (a)). Consequently, models trained on this dataset may rely on the presence of colorful patches to classify samples as benign, resulting in biased predictions (Rieger et al., 2020).

We replicate this bias synthetically, allowing for more precise control. Specifically, we generate a training dataset in which all benign samples contain a colorful patch, while malignant ones do not. We train three $\mathcal{X}$-ResNet-50 (Hesse et al., 2021) – models designed for training with attribution priors – with different priors on this biased data and evaluate them on validation sets with varying bias configurations in Table 2. Without any prior/debiasing, the model overfits to the colorful patches and fails when no such patch is available or its association is inverted. Attribution maps (*cf.* Figure 7 (b) – no debiasing) confirm that the model is focusing on

*Table 2.* **Validation results for the ISIC dataset with varying biases.** We report the accuracy (average over 5 runs) for the validation split of the ISIC dataset with different biases. In the "training bias" setup, the train bias is replicated with all the benign samples containing colorful patches, while in the "inverse bias" setup, the malignant samples contain colorful patches, as indicated by $^*$. A model with no debiasing learns the dataset bias and fails to classify samples when the bias is *not* present. A model with presence debiasing (existing attribution priors) can reduce this bias; however, it still fails to classify malignant samples when inserting colorful patches, indicating that it is biased based on the *absence* of colorful patches. Our proposed presence+absence debiasing results in the highest average accuracy for both setups without the training bias, and is similarly performant as a model trained without bias, suggesting that the model is largely debiased. "Attr." shows the relative attribution within the colorful patches, confirming qualitative results from Figure 7.

| Training bias | Model | Validation split (training bias) | | | | Validation split (inverse bias) | | | | Validation split (no bias) | | |
|---|---|---|---|---|---|---|---|---|---|---|---|---|
| | | Benign$^*$ | Malignant | Avg. | Attr. | Benign | Malignant$^*$ | Avg. | Attr. | Benign | Malignant | Avg. |
| None | $\mathcal{X}$-ResNet-50 | – | – | – | – | – | – | – | – | 0.84 | 0.77 | **0.81** |
| Benign$^*$ | No debiasing | 1.00 | 0.99 | **0.99** | 0.40 | 0.04 | 0.00 | 0.02 | 0.47 | 0.04 | 0.99 | 0.51 |
| | Presence debiasing | 0.96 | 0.88 | 0.92 | 0.08 | 0.66 | 0.17 | 0.41 | 0.13 | 0.66 | 0.88 | 0.77 |
| | Presence+absence debiasing (ours) | 0.91 | 0.88 | 0.89 | **0.07** | 0.74 | 0.43 | **0.59** | **0.08** | 0.74 | 0.88 | **0.81** |

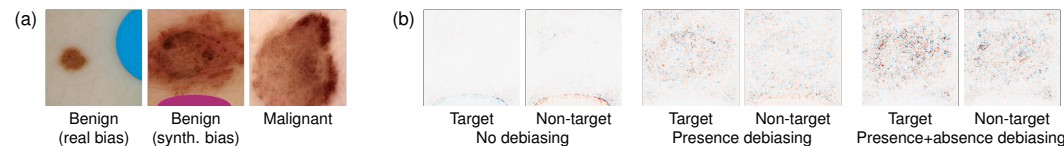

(a) Benign (real bias) — Benign (synth. bias) — Malignant
(b) Target / Non-target — No debiasing | Target / Non-target — Presence debiasing | Target / Non-target — Presence+absence debiasing

*Figure 7.* **Images and attributions for our biased ISIC dataset.** *(a)* We replicate the ISIC bias (real bias) that co-occurs with the benign samples with a synthetic bias. *(b)* Attributions of different (de)biased models for the benign sample with a synthetic bias. The target attribution is computed for the benign output logit and the non-target attribution for the malignant output. Only including absences in the debiasing prevents the model from relying on patch absence to predict malignancy.

the colorful patch.

To debias such a model, attribution priors (Ross et al., 2017; Rieger et al., 2020) have been proposed. Here, usually, the target attribution for each sample containing a spurious correlation is computed and constrained to be as low as possible in the area of the spurious correlation. When training with such an attribution prior (presence debiasing), the model performs well on validation data without bias, suggesting successful debiasing. However, when the bias is inverted (*i.e.*, benign samples lack colorful patches and malignant ones contain them), the accuracy drops significantly – particularly due to frequent misclassification of malignant samples. As argued in this work, the model may have learned to ignore the presence of colorful patches for benign predictions, yet still relies on their absence to predict malignancy – something not addressed by the attribution prior. This is further supported by the attribution maps in Figure 7 (b) – presence debiasing: the non-target (malignant) attribution for a benign sample with a colorful patch highlights the patch with negative attribution, indicating that it acts as an inhibitory signal for predicting malignancy.

We, therefore, propose *presence+absence debiasing*: extending the attribution prior to also include our proposed non-target attribution. This effectively suppresses patch attribution for the malignant output on benign samples and prevents the model from using either the presence or the absence of the colorful patch as a shortcut. As a result, we achieve a higher accuracy on the unbiased and inverted-bias validation sets, with attribution maps showing reduced

reliance on the patch across both classes (*cf.* Figure 7 (b) – presence+absence debiasing). Intriguingly, training and evaluating a model on unbiased data – which serves as an upper bound – achieves the same average accuracy as our proposed debiasing, indicating that our strategy successfully removes the bias. In Appendix B.4, we replicate this experiment with varying bias strength and a Vision Transformer (Dosovitskiy et al., 2021), yielding the same conclusions.[2]

## 6. Discussion

**Limitations and opportunities.** Non-target attribution requires computing more attribution maps than standard target attribution, which can limit scalability. For example, in the debiasing experiment in Section 5.4, it roughly doubled the computational cost, with overhead increasing further as the number of classes/concepts grows. To mitigate this, one can first use feature visualization through minimization to identify candidate encoded absences and then restrict non-target attribution to samples containing these concepts, as demonstrated in Appendix B.3; automatic concept extraction methods (*e.g.*, Rao et al., 2024) could further reduce redundancy. Additionally, as discussed

---

[2] Interestingly, Ross et al. (2017) observed that computing attributions for multiple classes slightly improved the stability of their attribution prior, which they attributed to discontinuities near decision boundaries. While conceptually similar to our presence+absence debiasing, they did not provide the theoretical insight or empirical analysis we develop here.

in Appendix A.2, our formulation in Definition 2.1 is based on an idealized intervention, whereas practical implementations rely on approximations. Consequently, to obtain more robust evidence, we complement our proposed feature visualization via minimization with non-target attributions, and, in Appendix B.5, further validate our findings using interventions aligned with the data-generation process. An interesting direction for future work is the study of more realistic interventions, for instance through counterfactual generation. Further, our experiments focus on image classification to isolate encoded absences; extending this analysis to more complex models or tasks is an important direction for future work. For example, in large-language models, non-target attributions could reveal inhibitory relationships between concepts that influence next-token predictions, and in image generation models, minimization-based feature visualization could help diagnose which visual concepts suppress certain generated attributes, enabling more precise control over model outputs. Finally, we here assume that concepts are axis-aligned with individual neurons, which is not necessarily true; in Appendix A.4 we outline why this assumption is tolerable and how to relax it.

**Conclusion.** In this work, we show that even concepts not present in the input can affect a neural network's output – a critical but largely overlooked aspect in XAI that often utilizes local explanations that only use information from the inputs under inspection. While mainstream explanation methods struggle to reveal such effects when applied in their standard form, simple modifications to attribution and feature visualization make these effects visible and complement existing explanations. Applying these tools to ImageNet models, we find that encoded absences are pervasively used, particularly for fine-grained classification. Moreover, we demonstrate that biases can arise not only from concept presence but also from concept absence, and that effective debiasing must account for both. While we only take initial steps in this direction, our findings suggest that encoded absences are not only common but fundamental to how models represent and use information. We hope this work opens the door to a broader rethinking of what constitutes an explanation in mainstream XAI.

## Impact Statement

Since our method is largely independent of any specific downstream application – such as image generation or facial recognition – we do not anticipate *direct* negative societal impact in such domains. However, our approach contributes to a deeper understanding of neural networks, which could carry *indirect* risks. For instance, improved insights into model behavior might enable the extraction of sensitive information from training data. Similarly, an enhanced mechanistic understanding could potentially be misused to manipulate models into producing targeted outputs, akin to adversarial attacks. Moreover, while we demonstrate how our method can be applied to debias models, it is conceivable that the same insights and techniques could be reversed to intentionally introduce bias.

While these risks are important to acknowledge, a more comprehensive understanding of model behavior also enables substantial positive societal impacts. These include the ability to debias models by reducing their reliance on sensitive or critical features, foster trust in machine learning systems, and identify and mitigate model vulnerabilities or limitations.

If our algorithm does not function as intended – for example, in models with symmetric activation functions where it is unclear if the absence of a feature is encoded in positive or negative activations (*cf.* Appendix A.4) – it could lead to incorrect interpretations and, consequently, incorrect adjustments applied to the model. It is, therefore, crucial to be aware of the theoretical limitations of the method and, in cases of uncertainty, to conduct deeper analyses of the model to ensure a correct understanding of how a given neuron behaves.

## Acknowledgments

RH and SR have received funding from the European Research Council (ERC) under the European Union's Horizon 2020 research and innovation programme (grant agreement No. 866008). SSM has been funded by the Deutsche Forschungsgemeinschaft (DFG, German Research Foundation) – project No. 529680848. Further, SR was supported by the DFG under Germany's Excellence Strategy (EXC 3066/1 "The Adaptive Mind," project No. 533717223). Additionally, SR and SSM have received funding from the DFG under Germany's Excellence Strategy (EXC-3057/1 "Reasonable Artificial Intelligence", Project No. 533677015). Moreover, JH has been funded by the Boehringer Ingelheim Foundation.

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

# A. Theoretical Elaborations

The main text focuses on presenting our core theoretical insights. Here, we provide additional elaborations to complement the main paper.

## A.1. Hassenstein-Reichardt detector

In the visual system of the fruit fly *Drosophila melanogaster* (Borst & Groschner, 2023), lobula plate tangential neurons are activated, for example, by rightward motion and inhibited by leftward motion, which ensures appropriate reactions to approaching predators, whose looming movement induces motion in multiple directions. Essential aspects of this computation are captured by the Hassenstein–Reichardt detector model, which computes global motion by subtracting the outputs of mirror-symmetric local motion detectors (Hassenstein & Reichardt, 1956; Reichardt, 1961; Egelhaaf et al., 1989; Haag et al., 2004) (*cf.* Figure 8). Consequently, the output neuron of the Hassenstein–Reichardt detector encodes the presence and absence of two concepts alike (rightward, resp. leftward motion).

## A.2. Implicit assumptions and approximations

For clarity and completeness, we make explicit several assumptions and approximations underlying our formulation and methods. These assumptions and approximations do not affect the validity of our arguments or empirical findings, but spelling them out helps to avoid potential ambiguities.

**Concept.** In our framework, and consistent with existing concept discovery methods, we define a concept as a set of input patterns, which we typically assume to be semantically related (*e.g.*, images of "dog snouts") for interpretability purposes. Strictly speaking, however, a concept in our formal Definition 2.1 can be any family of input patterns that share a common effect on a neuron – semantic coherence is a natural and practical assumption but not a formal requirement. In Appendix A.4, we further discuss how the framework extends to arbitrary feature space directions, which typically correspond to more semantically coherent concepts than individual neurons. Semantic interpretations (*e.g.*, the word "dog snout") are post-hoc descriptions of such families, rather than part of our formal definition itself. In our experiments, concepts are operationalized via feature visualization through minimization or image regions identified through negative attributions; in principle, more sophisticated approaches, such as CRAFT (Fel et al., 2023), could also be used to identify such pattern families. However, such methods would require appropriate adjustments to be compatible with encoded absences.

**Concept presence and fixed input context.** Definition 2.1 relies on the notion of a concept being present or absent within a fixed input context. In natural images, however, concepts cannot be perfectly inserted or removed without affecting the context. For example, removing a visual concept necessarily requires filling the affected region with other content, which may itself introduce new concepts. Consequently, this limits a perfectly precise notion of concept removal in the image domain and we can only approximate concept removal as defined in Definition 2.1. However, we did not find this to affect our theoretical arguments, proposed methods, or experimental results.

**Negative attributions as evidence of absence.** In Section 4, we identify encoded absences via negative attributions. This implicitly assumes that removing a concept highlighted by negative attribution would replace it with content that carries no information for the target prediction (*i.e.*, features with attribution close to zero), such that the inequality in Definition 2.1 is satisfied. While this replacement is not performed explicitly, our empirical results indicate that negative attributions reliably correspond to inhibitory evidence under this assumption.

**Input context in feature visualization through minimization.** Similarly, feature visualization through minimization implicitly assumes a fixed input context given by the noisy initialization used during optimization, and treats the concept of interest as spanning the full input. Under this assumption, feature visualization through minimization satisfies the inequality in Definition 2.1. Importantly, we show in Section 5.3 that the resulting encoded absences generalize beyond this specific context: concepts identified via minimization also inhibit activation when inserted into highly activating natural images. This demonstrates that not relying on a fixed input context is tolerable and that encoded absences generalize beyond specific input contexts.

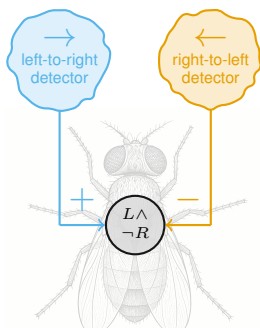

*Figure 8.* **Simplified illustration of the Hassenstein-Reichardt detector in *Drosophila*.** The activation of two subunits – encoding right-to-left ($R$) and left-to-right ($L$) movements – is subtracted. The output neuron encodes the *presence* of left-to-right movements while encoding the *absence* of right-to-left movements ($L \land \neg R$).

### A.3. Feature visualization and counterfactual explanations in the causal framework

In Section 2.1 of the main text, we view a DNN $f$ as a structural causal model $\mathfrak{C}$ and argue that the goal of XAI is to find a simplified causal model $\mathfrak{C}'$ that preserves task-relevant causal relationships while improving human interpretability (Hesse et al., 2023; Carloni et al., 2025). Here, we provide a simplified causal model $\mathfrak{C}'$ for two additional common XAI methods, feature visualization and counterfactual explanations.

For feature visualization (Olah et al., 2017), $\mathfrak{C}'$ reduces the model to a single causal path $z_j := f_j^{(1 \rightarrow l)}(x)$ from the input $x$ to a chosen internal neuron $z_j$ in layer $l$, and seeks the input $x = \tilde{x}$ that maximizes the positive activation of the intervention $do(x := \tilde{x})$ on $z_j$.

For a counterfactual explanation such as those of Goyal et al. (2019), $\mathfrak{C}'$ simplifies the model to capture the causal relationships necessary to identify the ("minimal") intervention $do(x := \overline{x})$ that changes the prediction from $y = f(x)$ to a desired counterfactual outcome $y' = f(\overline{x})$. This allows us to understand how an input would need to change to result in another prediction or to obtain the class-discriminative features in a sample.

### A.4. Alternative implementations for encoded absences

In Section 2.2 of the main paper, we outline a specific algorithm for encoding absences – inhibitory activation by the concept whose absence is encoded combined with a positive potential through another concept – that proved particularly relevant in our experimental setting. However, numerous alternatives could be considered, and we outline a few additional examples here.

The positive potential can be implemented in different ways as illustrated in Figure 9. The positive potential can not only be supplied via *(a)* using the activation of another concept $\tilde{x}$ in $l - 1$, but also via *(b)* a learned averaging over the previous layer (Hesse et al., 2021), or via *(c)* the bias term.

The above mechanistic processes work for unnormalized and ReLU (Fukushima, 1969) activations, as found in many image classification models. When relaxing these constraints, there are additional strategies to encode the absence of a concept $\hat{x}$. For example, instead of having a negative connection from a neuron in layer $l - 1$ encoding the presence of $\hat{x}$ to the neuron $z_j$ in layer $l$ encoding the absence of $\hat{x}$, there could be positive connections to all other neurons but $z_j$. After normalization, the presence of $\hat{x}$ leads to the inhibition of $z_j$, thereby satisfying the condition outlined in Definition 2.1. A neuron that is followed by a symmetric/unbounded activation function, such as Tanh or leaky ReLU, could encode the presence of a feature $\hat{x}$ in the positive direction and its absence in the opposite negative direction, requiring no positive potential. Interestingly, the model could even learn to encode the presence of a feature $\hat{x}$ in the *negative* direction and its absence in the opposite *positive* direction. We leave the identification of such cases to future work. However, once identified, feature visualization by maximization and our proposed feature visualization by minimization must be interpreted inversely to yield the intended explanations. Activation functions such as Swish (Ramachandran et al., 2018) and GeLU (Hendrycks & Gimpel, 2016), can be seen as smooth variants of ReLU, and thus, are capable of implementing the same mechanisms for encoded absences as outlined for ReLU activation functions. Since self-attention layers in Transformer architectures can be viewed as a generalization of linear layers, Vision Transformers can encode absences through similar mechanisms, as we empirically confirm in Appendices B.3 and B.4.

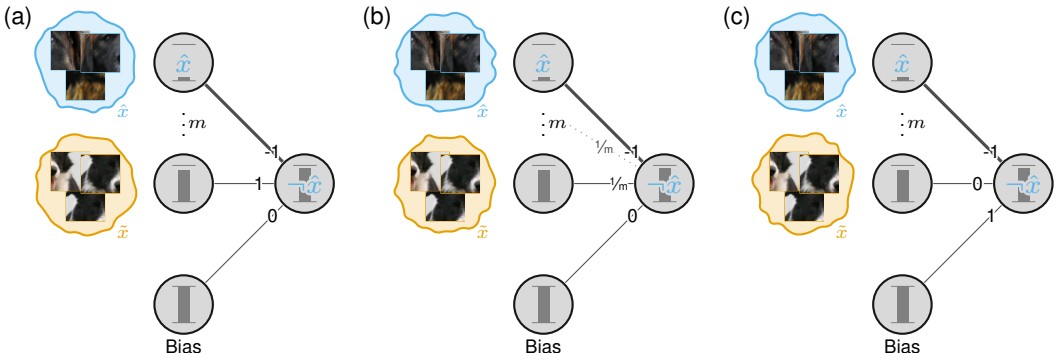

*Figure 9.* **Three mechanistic processes to encode the absence of a feature.** A neuron encoding the *absence* of concept $\hat{x}$ (*i.e.*, $\neg\hat{x}$) can be implemented by having a negative connection to a neuron encoding $\hat{x}$ and a positive potential through *(a)* another activating concept $\tilde{x}$, *(b)* some form of averaging, or *(c)* the bias.

So far, for simplicity, we have assumed that concepts are axis-aligned with individual neurons. In practice, however, concepts may lie along arbitrary directions in feature space (Elhage et al., 2022; O'Mahony et al., 2023), giving rise to *polysemantic* neurons. Fortunately, our proposed arguments and methods naturally extend to this case by substituting "neurons" with "feature space directions." Concretely, let $z^{(l)}(x) \in \mathbb{R}^{d_l}$ denote the (pre-activation) representation at layer $l$, and let $v \in \mathbb{R}^{d_l}$ be a unit vector defining a feature-space direction. We define the activation along $v$ as the inner product between $v$ and $z^{(l)}(x)$, i.e., $a_v(x) := \langle v, z^{(l)}(x) \rangle$. Then Definition 2.1 generalizes as follows:

**Definition A.1** (Encoded Absence for a Feature-Space Direction). If there exists a concept $\hat{x}$ such that its presence decreases the activation along direction $v$ in layer $l$, i.e.,

$$a_v([x, C_{\hat{x}} = 1]) < a_v([x, C_{\hat{x}} = 0]),$$

we say that the direction $v$ encodes the *absence* of $\hat{x}$ in the input context of $x$.

Similarly, in our proposed feature visualization through minimization, we could find input patterns that inhibit a specific feature space direction instead of a specific neuron.

The validity of our conclusions is not affected by polysemanticity. Polysemanticity would simply increase the complexity of what a channel encodes. Instead of representing the presence of concepts from one class and the absence of concepts from related classes, as shown in Section 5.3, a polysemantic channel could additionally encode the presence or absence of other, (un)related concepts. This would enrich the interpretation but does not undermine the conclusions we draw.

Please note that finding such meaningful feature space directions is an active area of research (Kim et al., 2018; Fel et al., 2023; O'Mahony et al., 2023) and not the scope of this paper.

### A.5. Stability of the positive potential

For encoded absences to work, a positive potential must be provided, which could be challenging for sparse inputs. In the simplest case, when the positive potential is provided via a bias term, it is independent of the input and thus remains stable even under extreme sparsity. Beyond that, the positive potential can also arise from features that are consistently present across the data distribution (*e.g.*, background statistics or co-occurring features), providing a stable baseline activation (up to a certain level of sparsity). That said, our primary focus is on natural image data, where inputs are typically dense, and sparsity is less pronounced; a similar argument applies to language models. In contrast, for tabular data, where high sparsity is more common, inputs may explicitly encode both presence and absence, potentially reducing the need for implicit absence encoding.

## B. Experimental Details

In this section, we provide detailed information to facilitate the reproduction of our experiments described in Section 5. All experiments have been run on a single Nvidia A100-SXM4 (80GB) or Nvidia RTX A6000 (48GB) GPU and require

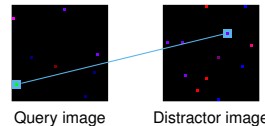

Query image      Distractor image

*Figure 10.* **Counterfactual visual explanation.** Counterfactual visual explanation (Goyal et al., 2019) for the two images shown in Figure 4 and the corresponding trained toy model. The explanation correctly highlights the green pixel (zoom in) as the concept whose swap induces a class change, thereby identifying it as the most class-discriminative concept. However, this explanation does not clarify whether the model relies on the presence of the green pixel, its absence, or both, which is possible with our proposed modifications.

only several hours ($\leq 10$) to complete. All code is implemented in PyTorch (Paszke et al., 2017) (3-Clause BSD license). To compute Integrated Gradients (Sundararajan et al., 2017) attributions (zero baseline) in Sections 5.1 to 5.3, we use Captum (Kokhlikyan et al., 2020) (3-Clause BSD license). Please refer to the main paper for an overview of each experiment and additional details.

### B.1. Explaining encoded absences in a Hassenstein-Reichardt detector

As illustrated in Figure 3 (b), we use a two-layer convolutional neural network with ReLU activation functions for the experiment introduced in Section 5.1. Each layer consists of two channels, with kernel sizes $(C = 2) \times (H = 1) \times (W = 2)$ and $(C = 2) \times (H = 1) \times (W = 1)$, respectively (no bias is used). Since we manually set the weights for the model (see Figure 3 (b) for exact weights), no training procedure is needed.

The non-target attribution is computed through $\mathcal{A}(x, t', f)$ for both visualized input samples $(x^{(1)}, t^{(1)})$ and $(x^{(2)}, t^{(2)})$, where $t'$ is the complementary class of $t$ in the binary classification setting.

### B.2. Explaining encoded absences in a trained toy model

For our toy experiment in Section 5.2, we generate a synthetic training dataset of $20\,000$ images of size $32 \times 32$ containing 8–12 non-green pixels, half of which contain one additional green pixel. Non-green pixels are generated by randomly assigning values of 0, 0.5, or 1 to the red and blue channels, respectively, excluding pure black (*i.e.*, both channels set to zero). The testing dataset contains 1000 images generated in the same fashion. As illustrated in Figure 4 (b), we use a two-layer convolutional neural network with ReLU activation functions. Each layer consists of two channels, with kernel sizes $(C = 3) \times (H = 1) \times (W = 1)$ and $(C = 2) \times (H = 1) \times (W = 1)$, respectively (no bias is used). We train the model with a binary cross-entropy loss, using an Adam optimizer (Kingma & Ba, 2015) with a learning rate of 0.01 and weight decay of 0.0001; we train for 15 epochs with a batch size of 256. Since the model does not always converge reliably (probably due to its simplicity), we perform five independent training runs and report results based on the best-performing model.

The non-target attribution is computed through $\mathcal{A}(x, t', f)$ for both visualized input samples $(x^{(1)}, t^{(1)})$ and $(x^{(2)}, t^{(2)})$, where $t'$ is the complementary class of $t$ in the binary classification setting.

To empirically verify our argument in Section 3 that counterfactual explanations, when applied in their standard form, are not well suited for explaining encoded absences, we present an illustrative example. Specifically, we apply the counterfactual visual explanation method (Goyal et al., 2019) to the two input images shown in Figure 4 and the corresponding trained toy model. The method identifies patches between a query image from class 1 and a distractor image from class 2 such that swapping these patches changes the model's prediction for the query image to that of the distractor image. In other words, it finds the most class-discriminative patches between the two images. In Figure 10, we visualize the resulting counterfactual explanation. As expected, the counterfactual explanation highlights the green pixel in the query image, since this patch is the most class-discriminative and is sufficient to flip the model's prediction when swapped. However, this type of explanation does not clarify whether the model's decision relies on the presence of the green patch, its absence, or both. Consequently, this approach is not suited for explaining encoded absences at the same level of fidelity as our proposed modifications.

### B.3. Explaining encoded absences in image classification models

**Quantitative.** For our quantitative analysis of inhibitory signals in ImageNet-trained models, we use the ImageNet-1k validation split (Russakovsky et al., 2015) and PyTorch (Paszke et al., 2017) torchvision models (VGG19 (Simonyan & Zisserman, 2015), ResNet-50 (He et al., 2016), ViT-B/16 (Dosovitskiy et al., 2021)). For each channel/neuron in the last

*Table 3.* **Quantitative evaluation of encoded absences.** We report the CNN results from Table 1 alongside their standard deviations (indicated by "±") and under different hyperparameters (patch size and number of images). Please refer to Table 1 for a detailed description.

| Model | Patch size | Nr. images | None | +Random | +Logical NOT (Mu & Andreas, 2020) | +Most act. | +Least act. (ours) |
|---|---|---|---|---|---|---|---|
| VGG19 (Simonyan & Zisserman, 2015) | 32 | 100 | 2.98 ± 1.08 | 2.84 ± 1.09 | 2.70± 1.04 → 2.68± 1.04 | 3.39 ± 1.10 | 2.14 ± 1.13 |
| VGG19 | 48 | 100 | 2.98 ± 1.08 | 2.68 ± 1.10 | 2.53± 1.06 → 2.51± 1.06 | 3.88 ± 1.16 | 0.94 ± 1.16 |
| VGG19 | 64 | 100 | 2.98 ± 1.08 | 2.41 ± 1.12 | 2.25± 1.08 → 2.21± 1.08 | 4.38 ± 1.13 | -0.38 ± 1.18 |
| VGG19 | 48 | 50 | 3.72 ± 1.08 | 3.39 ± 1.10 | 3.21± 1.06 → 3.19± 1.06 | 4.59 ± 1.12 | 1.66 ± 1.16 |
| VGG19 | 48 | 200 | 2.25 ± 1.07 | 1.98 ± 1.10 | 1.85± 1.04 → 1.84± 1.05 | 3.18 ± 1.10 | 0.25 ± 1.14 |
| ResNet-50 (He et al., 2016) | 32 | 100 | 0.18 ± 0.06 | 0.17 ± 0.06 | 0.16± 0.06 → 0.16± 0.06 | 0.21 ± 0.06 | 0.12 ± 0.07 |
| ResNet-50 | 48 | 100 | 0.18 ± 0.06 | 0.16 ± 0.06 | 0.15± 0.06 → 0.15± 0.06 | 0.25 ± 0.07 | 0.03 ± 0.11 |
| ResNet-50 | 64 | 100 | 0.18 ± 0.06 | 0.14 ± 0.07 | 0.13± 0.07 → 0.14± 0.07 | 0.29 ± 0.08 | -0.09 ± 0.11 |
| ResNet-50 | 48 | 50 | 0.21 ± 0.05 | 0.19 ± 0.06 | 0.15± 0.06 → 0.15± 0.06 | 0.29 ± 0.07 | 0.06 ± 0.11 |
| ResNet-50 | 48 | 200 | 0.14 ± 0.06 | 0.12 ± 0.06 | 0.11± 0.06 → 0.11± 0.06 | 0.22 ± 0.07 | -0.01 ± 0.11 |

backbone layer, we identify the 100 images that most strongly activate the respective channel / neuron after global average pooling (GAP) / taking the minimum over tokens. As we found the CLS token of the ViT to be largely insensitive to local patch insertions, we instead use a localized measure based on token-level activations. For each sample, we aggregate by taking the minimum over tokens, which captures the strongest local suppression effect. To assess the effect of interventions, we modify each of these 100 images by inserting either a random $48 \times 48$ patch, a patch containing the concept of the logical NOT (Mu & Andreas, 2020), the most activating patches, or one of the eight least activating $48 \times 48$ patches into a randomly selected corner of the image. To find the least/most activating patches, we use a sliding-window approach with a stride of 16. For identifying logical NOTs from (Mu & Andreas, 2020), we use the default hyperparameters with the only exception of reducing the beam search limit to 50, which was recommended by the authors for getting good explanations in a reasonable time. We compute the average channel/neuron activation (after GAP/min) across all modified images and all channels/neurons. In Table 3, we report the mean activation values from Table 1 alongside the corresponding standard deviations for the CNN-based backbones. Since (Mu & Andreas, 2020) do not identify a logical NOT for every channel, we evaluate their method only on the subset of channels for which such a concept is found. For a fair comparison, for each of these channels, we again report the mean activations after inserting a random patch as well as after inserting a patch containing the identified logical-NOT concept (random patch → logical-NOT patch). We additionally test different hyperparameter configurations for the CNN-based models and observe the same pattern: in both models, there exist patches that inhibit the activation of specific channels, indicating that the models utilize encoded absences (*cf.* Section 5.3).

To test if encoded absences also appear in earlier layers, we replicate the experiment for the first three blocks of a ResNet-50, and report results in Table 4. Across all layers, we observe evidence of suppressive signals/encoded absences, as indicated by lower scores for "+ Least act." compared to the random baseline. However, this effect varies across layers: in the first and last blocks, the difference is pronounced (approximately factors of 3 and 5, respectively), while in intermediate blocks it is notably smaller. One possible explanation is that early layers capture low-level image statistics, where the absence of certain patterns can already provide a valuable signal, whereas intermediate layers primarily build features through presence-based reasoning. In later layers, the absence of these higher-level features may then become more relevant for final predictions.

We note that our layer-wise measurements are taken at the output of individual convolutional blocks before residual addition, whereas in the forward pass, these representations are combined with the skip pathway. This allows features, including encoded absences from earlier layers, to be preserved and integrated at later stages, and thus, the effect of encoded absences could be more pronounced when considering activations after skip connections. Generally, strong absence encoding in later layers does not necessarily require equally strong absence encoding in all preceding layers. As shown by our mechanistic construction in Section 2.2, absences can even be implemented in layer $L$ when layer $L - 1$ only encodes presences.

To better understand *how many* channels encode absences, we further measure the fraction of channels/neurons in the final layer of the analyzed models that are statistically significantly affected by an inhibitory effect (*i.e.*, where the activation of a channel differs between images with the least activating patch inserted and those with random patches). Remarkably, this holds for $512/512$ channels in VGG-19, $2036/2048$ channels in ResNet-50, and $615/768$ neurons in ViT-B/16. Thus, most channels/neurons encode absences, indicating that this phenomenon is a systematic property of image classification models and warrants further investigation.

**Qualitative.** To find the qualitative examples from Figure 6, we start by computing Integrated Gradients (Sundararajan et al., 2017) attributions for each output logit with respect to the last convolutional layer of the above ResNet-50 (He et al.,

*Table 4.* **Quantitative evaluation of encoded absences in different layers.** We report the results from Table 1 in different layers of a ResNet-50. Please refer to Table 1 for a detailed description.

| Model | Layer | None | +Random | +Least act. (ours) |
|---|---|---|---|---|
| ResNet-50 | layer4[2].conv3 | 0.18 | 0.16 | 0.03 |
| ResNet-50 | layer3[2].conv3 | 0.26 | 0.24 | 0.18 |
| ResNet-50 | layer2[2].conv3 | 0.47 | 0.45 | 0.38 |
| ResNet-50 | layer1[2].conv3 | 0.08 | 0.07 | 0.02 |

2016) trained on ImageNet, using all validation samples of the corresponding class. Other layers, besides the penultimate one, could also have been used – later layers are likely to capture more high-level semantic features and may therefore be better suited for our analysis. We discard negative attributions because, for now, we focus only on channels that positively contribute to class prediction – *i.e.*, channels whose presence is important for predicting the class. We then average the attributions across samples. Channels are considered *important* for a specific class if their relative attribution (*i.e.*, attribution divided by total class attribution) is at least $0.05$. For each channel that is important for a specific class, we obtain the most activating patches for images from that class to visualize the encoded *presence*, respectively, the positive potential. Now that we know that the channel is important for predicting the class of interest and which *presences* cause it to activate, we aim to find which absences it encodes. To this end, the least activating patches for that channel are extracted from the entire validation split. For both the most and least activating patches, we extract the eight most/least activating candidate patches. We then manually select a monosemantic subset of three patches for more interpretable visualizations. While this manual selection does not affect the validity of our conclusions, it may convey a more monosemantic impression than is accurate – additional concepts may be present among the full set of eight patches (see Figure 11) as was discussed as a limitation in the main paper.

To further validate that these minimally activating patches carry meaningful semantics from the model's perspective, and are not merely an artifact, we classify each patch using the same ResNet-50 under inspection. In all three groups, at least one minimally activating patch is assigned to a semantically related class: for channel 2026, a patch is classified as "eft" (amphibian), for channel 1470 as "German shepherd," and for channel 494 as "squirrel monkey." These predictions indicate that the patches indeed contain meaningful concepts that the model associates with related classes, supporting our interpretation that the channel encodes the *absence* of these concepts.

**Impact of encoded absences on predictions.** To test whether encoded absences also affect predictions and not only activations, we run an additional ImageNet experiment on all correctly classified validation images of the considered ResNet-50. For each image, we identify the most important penultimate-layer channel (via InputXGradient), retrieve its 8 least-activating patches, and paste them into border locations. As a control, we also insert 8 random patches at the same locations. This yields a flip rate of $84\%$ vs. $8\%$ (least-activating *vs.* random). Since least-activating patches can introduce presence-based evidence, we also consider a suppression-only setting, which ensures that no neuron's activation increases relative to the original image. Under this constraint, the flip rate remains at $8.2\%$ vs. $3.5\%$ (least-activating *vs.* random), indicating that absence-related (inhibitory) signals still materially contribute to predictions at ImageNet scale. We note that we use only the single most important channel for finding the 8 least-activating patches, making this a conservative estimate; using multiple channels would likely strengthen the effect.

**Scaling non-target attributions** Scaling non-target attributions to large-scale datasets such as ImageNet-1k can be infeasible. We thus demonstrate a naive approach to obtain non-target attributions nonetheless. Specifically, for each penultimate-layer channel of the considered ResNet-50, we select the 10 least-activating patches from ImageNet-1k (and their source images), and compute non-target Integrated Gradients (IG) only on this small subset, using 10 random images as a control. This avoids exhaustive attribution over the full dataset. If the selected images contain encoded absences, we expect stronger negative attributions than for random images. Indeed, the results show that the average negative attribution is substantially higher ($178.25$ *vs.* $112.44$), showing that meaningful non-target attributions can be recovered without quadratic cost. We note that the average negative attribution of random images is still substantial, as they may contain similar suppressive features and IG is noisy.

### B.4. Debiasing models based on encoded absences

For our debiasing experiment in Section 5.4, we use the ISIC 2020 dataset (Rotemberg et al., 2021, CC-BY-NC license) of skin lesion images. Since the dataset is heavily imbalanced, with more benign than malignant samples, we randomly subsample the splits to create balanced sets, resulting in a training dataset of 1168 samples and an evaluation dataset of 524

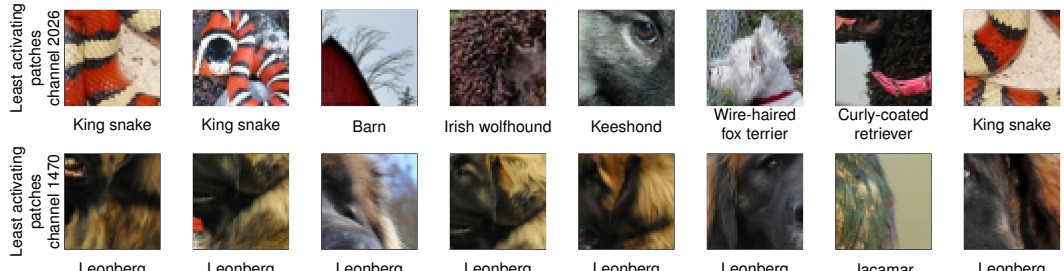

*Figure 11.* **The eight least activating patches for channels 2026 and 1470.** For the most and least activating patches in Section 5.3, we obtain eight candidate patches and manually select a monosemantic subset of three patches for more interpretable visualizations. Inspecting all eight patches for channels 2026 and 1470 in ResNet-50 reveals that these channels encode the absence of *multiple* concepts, consistent with prior work on polysemantic neurons (Elhage et al., 2022). The corresponding labels indicate the class of each patch.

samples. To increase the diversity of the samples, we apply random flipping and color jittering (brightness=0.2, contrast=0.2, saturation=0.2). The used $\mathcal{X}$-ResNet-50 model (Hesse et al., 2021) is pre-trained on ImageNet-1k (Russakovsky et al., 2015), with weights obtained from (Hesse et al., 2021, Apache-2.0 license). We finetune each model with a binary cross-entropy loss, using an Adam optimizer (Kingma & Ba, 2015) with a learning rate of 0.0001 and weight decay of 0.0001; we train for 20 epochs with a batch size of 128. The loss of the models with *no debiasing* on the unbiased and biased datasets can be written as

$$\mathcal{L} = \text{BCE}(x, t, f), \tag{1}$$

with BCE denoting the binary cross-entropy loss, $x$ the input sample, $t$ the target label, and $f$ the model. When training with *presence debiasing*, the loss becomes

$$\mathcal{L} = \text{BCE}(x, t, f) + 2\lambda \frac{|\mathcal{A}(x, t, f)\mathcal{P}(x)|}{|\mathcal{P}(x)| + 10^{-5}}, \tag{2}$$

with $\mathcal{A}(x, t, f)$ denoting the input attribution (Integrated Gradients; Sundararajan et al., 2017) and $\mathcal{P}(x)$ the segmentation mask of the colorful patch with 1 indicating its presence and 0 its absence; we dilate the mask by 10 pixels to include the edges. To prevent division by zero for images that contain no colorful patch, we add $10^{-5}$ to the denominator. We weight the attribution prior with a factor of 2 to account for the double attribution prior used in *presence+absence debiasing*, allowing for a fairer comparison. For *presence+absence debiasing*, the loss can be formulated as

$$\mathcal{L} = \text{BCE}(x, t, f) + \lambda \left( \frac{|\mathcal{A}(x, t, f)\mathcal{P}(x)|}{|\mathcal{P}(x)| + 10^{-5}} + \frac{|\mathcal{A}(x, t', f)\mathcal{P}(x)|}{|\mathcal{P}(x)| + 10^{-5}} \right), \tag{3}$$

with $t'$ being the complementary class of $t$ in our binary classification setting. In this experiment, only benign samples contain colorful patches during training, which means that the attribution prior for malignant samples is always zero ($|\mathcal{P}(x)| = 0$). Consequently, in all cases where the attribution prior has an effect, the true label $t$ corresponds to the benign class, and the complementary label $t'$ corresponds to the malignant class. Intuitively, in the *presence+absence debiasing* procedure, we compute the attribution for the malignant label on benign samples with colorful patches in order to assess the influence of these patches on malignant predictions. Each model is trained for 5 runs, and we determine the prior strength $\lambda \in \{1, 10, 100, 1000, 10000\}$ such that the resulting model performs the best on unbiased data. In Table 5, we expand on the results from Table 2 in terms of their standard deviations.

**Weaker bias.** To validate how our proposed debiasing behaves under weaker biases, we repeat the experiment in Section 5.4 with the bias being present in only 50% of the training samples, and report numbers in Table 6 (left). The model still learns the bias, albeit less strongly. Importantly, our presence+absence debiasing remains effective, particularly outperforming presence-only debiasing when the evaluation bias is inverted (inv. bias). This shows that addressing encoded absences is beneficial even when correlations are weaker.

**Vision Transformer (ViT).** To validate how our proposed debiasing behaves for different models, we repeat Section 5.4 with a ViT-B/16, and report numbers in Table 6 (right). The model also learns the bias and our presence+absence debiasing is the most effective for debiasing. Thus, ViTs are also capable of learning encoded absences and they require the same debiasing as CNNs.

*Table 5.* **Validation results for the ISIC dataset with varying biases.** We report the results from Table 2 together with their standard deviations (indicated by "±"). Please refer to Table 2 for a detailed description of the table.

| Training bias | Model | Validation split (training bias) | | | | Validation split (inverse bias) | | | | Validation split (no bias) | | |
|---|---|---|---|---|---|---|---|---|---|---|---|---|
| | | Benign* | Malignant | Avg. | Attr. | Benign | Malignant* | Avg. | Attr. | Benign | Malignant | Avg. |
| None | $\mathcal{X}$-ResNet-50 | – | – | – | – | – | – | – | – | 0.84 | 0.77 | **0.81** |
| | | – | – | – | – | – | – | – | – | ± 0.03 | ± 0.07 | ± 0.02 |
| Benign* | No debiasing | 1.00 | 0.99 | **0.99** | 0.40 | 0.04 | 0.00 | 0.02 | 0.47 | 0.04 | 0.99 | 0.51 |
| | | ± 0.00 | ± 0.02 | ± 0.01 | ± 0.02 | ± 0.02 | ± 0.00 | ± 0.01 | ± 0.02 | ± 0.02 | ± 0.02 | ± 0.00 |
| | Presence debiasing | 0.96 | 0.88 | 0.92 | 0.08 | 0.66 | 0.17 | 0.41 | 0.13 | 0.66 | 0.88 | 0.77 |
| | | ± 0.01 | ± 0.10 | ± 0.05 | ± 0.00 | ± 0.08 | ± 0.05 | ± 0.03 | ± 0.05 | ± 0.08 | ± 0.10 | ± 0.01 |
| | Presence+absence debiasing (ours) | 0.91 | 0.88 | 0.89 | **0.07** | 0.74 | 0.43 | **0.59** | **0.08** | 0.74 | 0.88 | **0.81** |
| | | ± 0.06 | ± 0.01 | ± 0.03 | ± 0.01 | ± 0.06 | ± 0.06 | ± 0.03 | ± 0.01 | ± 0.06 | ± 0.01 | ± 0.03 |

*Table 6.* **Left: Weaker bias.** We report the main results for the experimental setup from Section 5.4 when the bias is only present in 50% of the training samples. **Right: ViT.** We report the main results for the experimental setup from Section 5.4 for a ViT-B/16.

| Model | Train bias (Avg.) | Inv. bias (Avg.) | No bias (Avg.) |
|---|---|---|---|
| No debiasing | 0.93 | 0.41 | 0.82 |
| Presence debiasing | 0.92 | 0.46 | 0.83 |
| Pres.+abs. debiasing | 0.90 | 0.64 | 0.83 |

| Model | Train bias (Avg.) | Inv. bias (Avg.) | No bias (Avg.) |
|---|---|---|---|
| No debiasing | 0.96 | 0.16 | 0.62 |
| Presence debiasing | 0.80 | 0.56 | 0.71 |
| Pres.+abs. debiasing | 0.81 | 0.60 | 0.74 |

## B.5. Controlled concept intervention

As outlined in Section 6 and Appendix A.2, our formulation in Definition 2.1 is based on an idealized intervention, whereas practical implementations rely on approximations. For example, negative non-target attributions do not in general guarantee a causal encoded absence, which could in principle affect our experimental conclusions. To verify that this issue does not influence our findings, we conduct a qualitative analysis in Figure 12, comparing attributions for the same sample before and after introducing the encoded absence.

For each experiment from Sections 5.1, 5.2 and 5.4 that relies on non-target attributions, we take a sample from the class where the model is hypothesized to rely on an encoded absence: class 1 containing a left-to-right movement in the Hassenstein–Reichardt detector, class 2 containing no green pixel in the toy example, and a malignant sample containing no colorful patch in the debiasing experiment (left sample of each experiment in Figure 12). For each such sample, we compute the *target* attribution (shown below the respective sample in Figure 12). These attributions exhibit no strong negative regions (no strong red highlights), indicating that no or only minimal inhibitory signals are present in the unmodified inputs where the concept of the encoded absence is absent.

We then insert the concept corresponding to the encoded absence into the same samples (right sample of each experiment in Figure 12). Specifically, we include a right-to-left movement in the Hassenstein-Reichardt detector sample, a green pixel in the sample from the toy experiment (bottom left quarter; zoom in), and a colorful patch in the malignant sample from the debiasing experiment (bottom). We again compute the target attribution on these samples (shown below the respective sample in Figure 12). The inserted concepts produce substantially stronger negative attributions (red) than observed in the original samples. This confirms that the negative attributions used in our experimental sections genuinely arise from encoded absences rather than from unrelated effects.

Note that in this controlled setup, we use *target* attributions rather than non-target attributions. This is because the intervention explicitly inserts the concept whose absence is encoded into a sample from the class of interest, turning the input into one that now contains that concept. In such a setting, where the concept is present by construction, target attributions are the appropriate choice. In typical real-world scenarios and in our main experiments, we do not have access to such explicit concept insertions. The class of interest usually does not contain the concept whose absence is encoded, and the goal is precisely to detect how the model responds to that absence. As a result, non-target attributions must be computed on samples from *other* classes that do contain the concept, enabling us to identify the inhibitory relationship.

**Quantitative evaluation.** We further evaluate quantitatively whether our methodological approximations of Definition 2.1 align with the data-generation process. To this end, we compare the inhibitory effects obtained through patch-based interventions (as in Section 5.3) with those obtained from controlled interventions following the data-generating process (for the toy and ISIC setups in Sections 5.2 and 5.4). For both setups, we consider the output neuron known to encode an absence, take its corresponding class images, and include the hypothesized concept whose absence is encoded by the neuron.

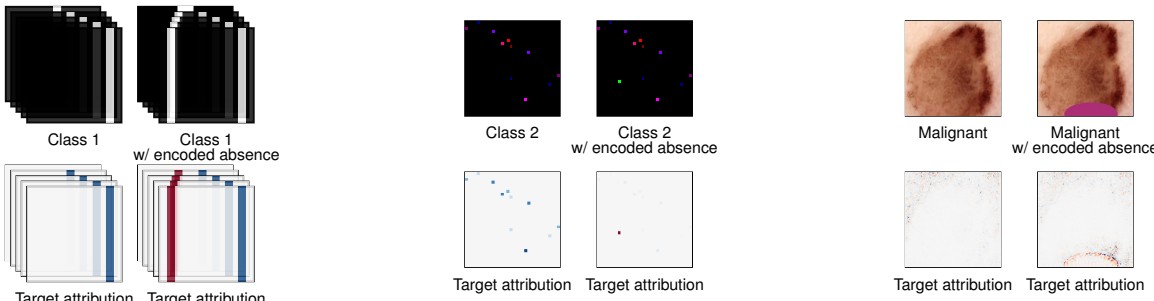

*Figure 12.* **Controlled concept intervention.** For each experiment relying on non-target attributions, we select a sample from the class that is hypothesized to use an encoded absence (left-side sample of each group). The target attributions for these unmodified samples show no strong inhibitory signals, as indicated by the absence of pronounced red highlights. After inserting the corresponding absence concept into the right-side samples (a right-to-left motion pattern in the Hassenstein–Reichardt detector example; a green pixel in the toy example, bottom-left quadrant; and a colorful patch in the malignant sample from the debiasing experiment) and recomputing the target attribution (see Appendix B.5 for details), we observe markedly stronger inhibitory responses, visible as strong red attributions. This controlled intervention provides clear evidence that the inhibitory patterns identified in our experiments indeed reflect encoded absences.

*Table 7.* **Quantitative evaluation of our intervention approximations.** Average activation and negative attribution values across datasets and intervention settings.

| Dataset | Original sample without intervention | | w/ controlled intervention | | w/ patch intervention | |
|---|---|---|---|---|---|---|
| | Avg. activation | Avg. neg. attr. | Avg. activation | Avg. neg. attr. | Avg. activation | Avg. neg. attr. |
| Toy | 0.22 | 0 | -0.11 | 10.44 | -0.11 | 10.43 |
| ISIC | -0.06 | 926.55 | -0.73 | 1469.09 | -0.98 | 1501.5 |

We further report the corresponding negative attributions to verify that attribution signals behave consistently under these interventions. Results are reported in Table 7 and confirm that controlled and patch-based interventions lead to similar and consistent suppression effects, and that the magnitude of negative attributions increases accordingly, as expected. Overall, while all interventions remain approximations of the ideal do-intervention (cf. Appendix A.2), their consistent behavior across different settings provides empirical support for the causal interpretation in Definition 2.1.

