# OpenReview forum: "What is Missing? Explaining Neurons Activated by Absent Concepts"
_ICML.cc/2026/Conference — ICML 2026 regular_

### Official Review · Reviewer_jAf9 · 2026-03-03

**Soundness:** 3
**Presentation:** 3
**Significance:** 3
**Originality:** 3
**Overall Recommendation:** 4
**Confidence:** 3

**Summary:**

The paper identifies "encoded absences" as a critical yet often overlooked causal relationship in deep neural networks (DNNs). Encoded absences are defined as instances in which the absence of a concept increases a neuron's activation. The authors argue that mainstream explainable AI (XAI) methods such as target attribution and feature visualization via maximization implicitly assume that relevant information resides in present features. This causes these methods to struggle to reveal absent concepts. To address this blind spot, the authors propose two straightforward extensions: non-target attribution and feature visualization through minimization. Through experiments involving simple architectures, ImageNet models, and a skin lesion debiasing task, the authors demonstrate that DNNs use absent concepts for fine-grained classification and that addressing these absences can improve model debiasing.

**Compliance With Llm Reviewing Policy:**

Affirmed.

**Final Justification:**

This paper provides a clear causal logical definition for the concept of "encoded absence", offers an existence proof from the perspective of neuronal connection mechanisms, and candidly discusses its limitations. The author's response addressed my concerns. Although there are still some unclear conceptual issues in the paper, based on the comments of other reviewers, I have decided to maintain my score.

**Key Questions For Authors:**

1. Architectural Scope and Generality

The empirical evidence in Section 5.3 regarding real-world models focuses exclusively on the final convolutional layers of CNN architectures, specifically VGG19 and ResNet-50. Have you investigated whether this phenomenon is specific to the inductive biases of convolutions or whether encoded absences appear in other architectures, such as Vision Transformers? Additionally, are these absences encoded in earlier, lower-level layers of the network?

2. Ambiguity of "Concept"

The manuscript frequently discusses the encoding of "concepts." However, feature visualization through minimization relies heavily on inserting localized 48×48 patches. How do you bridge the semantic gap between a high-level, human-interpretable "concept" and a localized pixel patch? Can you assure us that these minimally activating patches consistently map to coherent semantic concepts rather than out-of-distribution adversarial artifacts?

3. Causal Formulation and Interventions

The causal formulation introduces a binary variable, C_(hat x) \in \{0, 1\}$ to indicate the presence or absence of a concept. However, as briefly acknowledged in the appendix, a concept in a natural image is rarely an independent variable that can be intervened upon perfectly without altering the surrounding context. For example, occluding a concept introduces new edges and background content. How does this lack of interventional independence affect the purity of the causal claims made in the main text?

4. Gradients as Causal Effects

Expanding on the causal formulation, the paper relies on attribution methods, which are often gradient-based, to identify inhibitory signals. What explicit theoretical assumptions are required to justify the leap from a gradient-based attribution score to a true causal effect within your proposed structural causal model?

**Limitations:**

The authors include a dedicated "Discussion" section (Section 6), in which they explicitly acknowledge the limitations of scalability of non-target attribution and the current focus on image classification. They also provide an "Impact Statement" that addresses the potential dual-use risks of a better understanding of model behavior, such as extracting sensitive information or crafting adversarial attacks, while balancing these risks against the benefits of improved debiasing.

**Strengths And Weaknesses:**

Strengths:

1. Clear Motivation and Problem Definition: The paper identifies a fundamental blind spot in the current XAI literature: traditional target-only explanations inherently overlook absence-based evidence.

2. Simplicity and Effectiveness: The proposed methodological extensions, non-target attribution and minimization, are algorithmically simple yet highly effective solutions to the described problem.

3. Convincing Mechanistic Grounding: The theoretical explanation of how DNNs can implement these absences by relying on a negative connection to the absent concept and a positive potential from another source is highly persuasive. The toy experiments clearly demonstrate this mechanism.

Weaknesses:

1. Evaluation Methodology: In the ImageNet quantitative evaluation, the authors assess inhibitory effects by inserting patches into random corners of images and measuring the activation drop. This hard-insertion method could introduce out-of-distribution artifacts that might suppress activation for reasons unrelated to semantic absence. Although they use random patches as a baseline control to mitigate this, the evaluation remains somewhat artificial.

2. Scalability constraints: As the authors acknowledge in their discussion, computing non-target attributions scales poorly. It requires computing gradients for multiple potential absent classes, which roughly doubled the computational cost in their debiasing experiment and poses a bottleneck for datasets with thousands of classes.

3. Architectural Scope and Generality: The empirical evidence on real-world models (Section 5.3) focuses heavily on the final convolutional layer of CNN architectures (VGG19 and ResNet-50). It remains unclear if this phenomenon is specific to the inductive biases of convolutions or if it applies equally to other architectures (like Vision Transformers) and earlier, lower-level layers.

4. Ambiguity of "Concept": The paper frequently discusses "concepts," yet the application of feature visualization through minimization relies heavily on $48 \times 48$ patch-level insertions. There is a semantic gap between a high-level "concept" and a localized patch that is not fully reconciled in the text.

5. Causal Formulation Gaps: The paper introduces a causal variable $C_{\hat{x}} \in \{0, 1\}$ to indicate concept presence. However, in natural images, a concept is rarely an independent variable that can be intervened upon without altering the surrounding context. Furthermore, the leap from gradient-based attribution to true causal effect requires stronger assumptions that are not rigorously addressed in the main text.

---

> ### Author Rebuttal · Authors · 2026-03-30
>
> We thank the reviewer for the constructive feedback and address the key questions below:
>
> - **Q1: Architectural Scope and Generality.** We agree that including an analysis on ViTs strengthens our manuscript. Accordingly, in our reply to Q3 of Reviewer RGDo, we include results for the experiments from Secs. 5.3 and 5.4 for ViTs. In both cases, we find strong evidence that ViTs also encode absences and, in the case of Sec. 5.4, benefit from our proposed presence+absence debiasing.
>
>    Regarding lower-level layers, we replicate the experiment from Sec. 5.3 for the first three blocks of a ResNet-50. Results are shown in the table below. Across all layers, we observe evidence of suppressive signals/encoded absences, as indicated by lower scores for "+ Least act." compared to the random baseline. However, this effect varies across layers: in the first and last blocks, the difference is pronounced (approximately factors of 3 and 5, respectively), while in intermediate blocks it is notably smaller.
>
>    One possible explanation is that early layers capture low-level image statistics, where the absence of certain patterns can already provide a valuable signal, whereas intermediate layers primarily build features through presence-based reasoning. In later layers, the absence of these higher-level features may then become more relevant for final predictions. We will include this analysis in the revision.
>
>    |layer|None|+Random|+Least act.|
>    |---|---|---|---|
>    |layer4[2].conv3|0.18|0.16|0.03|
>    |layer3[2].conv3|0.26|0.24|0.18|
>    |layer2[2].conv3|0.47|0.45|0.38|
>    |layer1[2].conv3|0.08|0.07|0.02|
>
>
> - **Q2: Ambiguity of "Concept".** We agree that the abstraction gap between high-level concepts and localized patches is evident in our as well as previous work. Consistent with prior XAI work, in our framework, "concepts" are operationalized as input patterns that influence neuron activations; minimally activating patches are instances of such patterns rather than complete high-level, human-interpretable concepts. Such patch-based analyses are common in concept-based interpretability (e.g., CRAFT (Fel, 2023) uses localized patches), indicating that meaningful structure can be captured despite locality.
>
>    Importantly, in Sec. 5.3, these patches are not generated via optimization but are extracted from real images, making adversarial artifacts unlikely. We further control for out-of-distribution effects via random patch insertions. In practice, as shown qualitatively in Sec. 5.3 (e.g., Fig. 6), the identified patterns often correspond to semantically meaningful features from related classes. We will clarify this interpretation and more clearly distinguish between semantic and operational notions of "concept" in the revision.
>
> - **Q3: Causal Formulation and Interventions.** We refer to our response to Reviewer 6gdg (Q2) for a more detailed discussion. In brief, we acknowledge that patch-based interventions are only approximations and may introduce artifacts. In Sec. 5.3 (ImageNet), we use random patch insertions as a control to partially account for boundary and distributional effects, but agree that this might not fully resolve the issue.
>
>    To strengthen the ImageNet analysis, we additionally evaluate non-target attributions (see reply to Reviewer mF5n, Q5) and find that images from which the least-activating patches are drawn exhibit significantly more negative attribution than random images. This provides evidence for encoded absences that does not rely on insertion-based interventions.
>
>    We will clarify this multi-evidence perspective and its assumptions more explicitly in the revision.
>
> - **Q4: Gradients as Causal Effects.** We agree that attribution methods provide only approximations and do not, by themselves, establish causal effects. Importantly, this is a general limitation of attribution-based XAI: gradient-based attributions are widely used as approximations of causal influence, without a fully rigorous causal justification, even for presence-based explanations.
>
>    Our contribution is therefore orthogonal: we show that, under these same commonly used assumptions, absence-based signals are systematically missed unless non-target settings or minimization are considered.
>
>    To strengthen the connection to our causal formulation, we complement attribution-based analysis with approximate interventions in Sec. B.5. Specifically, we analyze attributions before and after introducing the encoded absence and observe consistent inhibitory (negative) attribution patterns, indicating that the identified signals are stable under intervention and not artifacts of the attribution method alone.
>
>    While this does not constitute a perfect causal test, it provides empirical support that attribution-based and intervention-based signals are aligned in our setting. We will clarify this distinction between attribution-based evidence and causal interpretation more explicitly in the revision.

---

> > ### Author Rebuttal · Reviewer_jAf9 · 2026-04-03
> >
> > The author has already answered my questions. I will maintain my score.

---

> > > ### Author Response · Authors · 2026-04-05
> > >
> > > We again thank the reviewer for their valuable feedback and are happy to have clarified all concerns.

---

### Official Review · Reviewer_RGDo · 2026-03-11

**Soundness:** 3
**Presentation:** 2
**Significance:** 3
**Originality:** 2
**Overall Recommendation:** 4
**Confidence:** 3

**Summary:**

This paper argues that mainstream XAI mainly explains concept presence, while often missing cases where a model relies on concept absence. It formalizes encoded absence as an interventional relationship and proposes two simple modifications of existing tools: Non-target attribution and feature visualization through minimization. Through toy experiments, ImageNet analysis, and a debiasing setup, the paper argues that absence-based evidence is practically relevant and can complement standard explanations.

**Compliance With Llm Reviewing Policy:**

Affirmed.

**Final Justification:**

The paper has a clear and meaningful conceptual contribution by arguing that concept absence, not only concept presence, can be an important part of model reasoning and explanation. During the rebuttal, the authors have addressed major comments and I believe that the paper has merits to be discussed in the conference. I have increased the score.

**Key Questions For Authors:**

• The paper would benefit from a clearer discussion of concept-based explanation methods such as CRAFT, CLIP-Dissect, and WWW. These methods can provide concept-level interpretations with signed relevance or contribution scores. The manuscript should clarify why such negative concept contributions are insufficient to establish encoded absence, and more explicitly distinguish correlational or decomposition-based negative evidence from the stronger causal notion introduced here.
• The debiasing result seems interesting and effective in the synthetic patch-bias setup. Would the proposed presence+absence debiasing strategy remain effective in more realistic settings with weaker or more distributed spurious correlations?
• The current experiments focus mainly on CNN-based vision models. Do the authors expect encoded absences to arise similarly in transformer-based or other attention-based vision architectures, and could this be validated empirically?

**Limitations:**

Yes

**Strengths And Weaknesses:**

Strengths
- The paper presents a clear and meaningful conceptual contribution by arguing that concept absence, not only concept presence, can be an important part of model reasoning and explanation.
- The proposed perspective is well motivated both theoretically and empirically, with controlled toy experiments and practical debiasing results supporting the main message.
- The method is simple and usable in practice, and the paper communicates its contribution clearly as a conceptual extension of existing XAI tools.

Weaknesses
- The paper has limited algorithmic novelty, but this seems acceptable given that the main intended contribution is the conceptual framing and its implications for XAI.
- The positioning against prior concept-based interpretability work is not fully developed, especially for methods that can already yield negative concept-level contributions, such as CRAFT [1*], CLIP-Dissect [2*], WWW [3*] and related neuron-description or concept-based interpretation frameworks. The comparison to prior concept-based interpretability methods could be sharpened further to better clarify the novelty and scope of the proposed idea.
- The empirical evaluation is concentrated on image classification, so the broader generality of the proposed perspective across other model families would benefit the result of the paper.


[1*] Fel, Thomas, et al. "Craft: Concept recursive activation factorization for explainability." CVPR 2023.
[2*] Oikarinen, Tuomas, and Tsui-Wei Weng. "Clip-dissect: Automatic description of neuron representations in deep vision networks." ICLR 2023.
[3*] Ahn, Yong Hyun, Hyeon Bae Kim, and Seong Tae Kim. "Www: a unified framework for explaining what where and why of neural networks by interpretation of neuron concepts." CVPR 2024.

---

> ### Author Rebuttal · Authors · 2026-03-30
>
> We thank the reviewer for the constructive feedback and address the key questions below:
>
> - **Q1: Concept-based explanation methods.** We agree that this discussion could be stronger. Our goal differs from these methods in one key respect: we focus on inhibitory relationships, i.e., whether the presence of a concept suppresses a neuron according to Definition 2.1, whereas most concept-based methods in their standard pipelines are designed to identify concepts associated with high activation or positive relevance. In that sense, these methods are naturally aligned with encoded presences, while encoded absences require an explicit shift toward minimally activating or inhibitory signals.
>
>    Concretely:
>
>    - **CRAFT** builds concepts for one class only from images that have been predicted as that class (see their Sec. 3.1), which emphasizes present evidence and makes absent-but-relevant concepts less likely to be surfaced directly.
>    - In **CLIP-Dissect**, a concept is assigned to a neuron if the neuron is highly active on images for which CLIP assigns a high similarity to that concept (their Sec. 3). For a neuron encoding the absence of a concept, however, the relevant signal lies in *reduced* activation when the concept is present, which would manifest as anti-correlation. While such anti-correlation may exist, it is not explicitly modeled or interpreted in their similarity measures (Sec. 3.2). This is further reflected in their WPMI similarity, which relies on the *most activating images*, as well as in their Fig. 1, where only the *top-activating samples* are visualized.
>    - Similarly, in **WWW**, only the highest activating images/crops are used (see, e.g., their Fig. 1).
>
>    Our point is therefore not that these methods are incompatible with our perspective, but that they are not designed to directly test encoded absences in their standard form – extending them to encoded absences could be a promising future direction. We will revise the related-work section to make this distinction more explicit.
>
> - **Q2: Presence+absence debiasing under weaker spurious correlations.** From our perspective, biases from encoded absences behave similarly to those from presences: models can learn weaker or more distributed correlations, though effects may decrease as correlations become less frequent.
>
>    To validate this, we repeat Sec. 5.4 with a weaker bias (present in only 50% of training samples). The model still learns the bias, albeit less strongly. Importantly, our presence+absence debiasing remains effective and consistently outperforms presence-only debiasing. This shows that addressing encoded absences is beneficial even when correlations are weaker. Results are shown below and will be included in the revision.
>
>    |Model|Train bias (Avg.)|Inv bias (Avg.)|No bias (Avg.)|
>    |---|---:|---:|---:|
>    |No debias|**0.93**|0.41|0.82|
>    |Presence|0.92|0.46|**0.83**|
>    |Pres.+Abs. (ours)|0.9|**0.64**|**0.83**|
>
> - **Q3: Encoded absences in Vision Transformers (ViTs).** As we show that encoded absences are generally useful for classification, we expect that ViTs also utilize mechanisms akin to encoded absences, given their sufficient degree of expressiveness.
>
>    We verify this empirically by replicating Sec. 5.3 and 5.4 for ViTs and find consistent evidence of encoded absences.
>
>    - **5.3 with ViT.** ​​As we found the CLS token to be largely insensitive to local patch insertions, we instead use a localized measure based on token-level activations. For each sample, we compute the change in activation per token and aggregate by taking the minimum over tokens, which captures the strongest local suppression effect. This differs from Sec. 5.3 for CNNs, where spatial aggregation is done through global average pooling.
>
>       As a result, values can be negative even in the "None" setting, since the minimum token activation can already lie below the reference level. While random patches induce some decrease, the least-activating patches consistently cause a larger drop, indicating structured suppressive signals (encoded absences) also in ViTs.
>
>       |Method|vit_b_16|vit_b_32|
>       |---|---:|---:|
>       |None|-0.14|-0.11|
>       |+ Rand|-0.18|-0.22|
>       |+ Least act.|-0.24|-0.34|
>
>
>    - **5.4 with ViT.** In the debiasing experiment, the ViT (vit_b_16) model without debiasing relies on the encoded absence bias, as shown by a strong performance drop under inverse bias (see results below). Our presence+absence debiasing achieves the best accuracy on both inverse-bias and unbiased splits, outperforming the presence-only debiasing approach.
>
>       |Model|Train bias (Avg.)|Inv. bias (Avg.)|No bias (Avg.)|
>       |---|---:|---:|---:|
>       |No debias|**0.96**|0.16|0.62|
>       |Presence|0.8|0.56|0.71|
>       |Pres.+Abs. (ours)|0.81|**0.6**|**0.74**|
>
>    This analysis suggests that ViTs can also learn to encode absences, and we thank the reviewer for this valuable suggestion that will be included in the revision.

---

> > ### Author Rebuttal · Reviewer_RGDo · 2026-04-02
> >
> > I have read other reviewer's comments and the author's rebuttal. Thanks for addressing major concerns!

---

> > > ### Author Response · Authors · 2026-04-05
> > >
> > > We again thank the reviewer for their valuable feedback and are happy to have clarified all concerns.

---

### Official Review · Reviewer_6gdg · 2026-03-11

**Soundness:** 3
**Presentation:** 3
**Significance:** 4
**Originality:** 4
**Overall Recommendation:** 5
**Confidence:** 3

**Summary:**

This paper explores a long-overlooked problem in the field of Explainable AI (XAI): Encoded Absences. This is a phenomenon where the activation of a neuron is not caused by the "presence" of a concept, but rather by its "absence".

**Compliance With Llm Reviewing Policy:**

Affirmed.

**Final Justification:**

I appreciate the authors' efforts in addressing my comments.

Since my primary concerns are now resolved, I will slightly increasing my score.

**Key Questions For Authors:**

1.The article mentions that implementing absence detection requires a source of "positive potential," which can be a bias term or other persistent features. However, whether this positive potential remains stable in extremely sparse inputs is less explored in the paper.

2.In the formula $$f_{j}^{(l)}(do(x:=[x,C_{\hat{x}}=1])) < f_{j}^{(l)}(do(x:=[x,C_{\hat{x}}=0]))$$, while the definition is clear, how can a perfect $do$-intervention (i.e., inserting a concept without changing any other variables) be achieved in natural image experiments? In natural images, inserting a concept without altering the background context seems difficult to implement. Therefore, is the observed decrease in activation merely due to disrupting the original texture (out-of-distribution) rather than semantic "concept inhibition"?

3.The current mechanistic proof is primarily based on the combination of ReLU activation functions and negative weights. Whether this theory remains robust for models using other activation functions (e.g., Swish, GeLU) or more complex structures (e.g., attention mechanisms in Transformers) requires more mathematical derivation or experimental support.

**Limitations:**

yes

**Strengths And Weaknesses:**

1.Soundness: The paper provides a clear causal logic definition for "encoded absence" and offers an existence proof from the perspective of neuronal connection mechanisms. This perspective, cutting in from Structural Causal Models, is very solid. The authors candidly discuss limitations, such as the computational overhead of non-target attribution and the difficulty of perfectly "intervening" on a concept in natural images.

2.Presentation: The paper's logic is rigorous ; it provides a good review of related works and clearly defines the boundary between its contributions and existing methods.

3.Significance: The paper demonstrates that traditional debiasing methods fail when dealing with biases triggered by "absences" and proposes effective improvement strategies. As a "patch" for existing tools, this method can be directly integrated into current explanation libraries. Its insights into fine-grained classification are highly instructive.

4.Originality: The paper systematically studies how DNNs utilize "absent concepts" for reasoning. It proposes a new Attribution Prior that combines both "presence" and "absence," which can more thoroughly remove spurious correlation biases in models.

---

> ### Author Rebuttal · Authors · 2026-03-30
>
> We thank the reviewer for their constructive feedback and for considering our reasoning rigorous. We answer the key questions raised below:
>
> - **Q1: Stability of the positive potential for sparse inputs.** In the simplest case, when the positive potential is provided via a bias term, it is independent of the input and thus remains stable even under extreme sparsity. Beyond that, the positive potential can also arise from features that are consistently present across the data distribution (e.g., background statistics or co-occurring features), providing a stable baseline activation (up to a certain level of sparsity).
>
>    That said, our primary focus is on natural image data, where inputs are typically dense, and sparsity is less pronounced; a similar argument applies to language models. In contrast, for tabular data, where high sparsity is more common, inputs may explicitly encode both presence and absence, potentially reducing the need for implicit absence encoding. We will clarify this role of sparsity and domain-dependent behavior in the revision.
>
> - **Q2: Perfect do-intervention in images.** We fully agree that perfectly isolating a concept via a do-intervention in natural images is not feasible. As discussed in Appendix A.2 (L.639–645), our formulation relies on an idealized intervention, while all practical implementations necessarily introduce approximations.
>
>    For this reason, our empirical argument does not rely on patch insertion alone. In Section 5.3, patch insertion is paired with random-patch controls to partly separate semantic suppression from generic perturbation effects; the random patches have only a minor effect, whereas the least-activating patches induce much stronger suppression. The paper and rebuttal (e.g., Q5 of Reviewer mF5n) also complement this with attribution- and minimization-based evidence, and Appendix B.5 includes a controlled intervention analysis showing that the relevant negative attribution patterns emerge after introducing the encoded absence. We will make this distinction between idealized causality and practical approximation more explicit in the revision.
> - **Q3: Extension to other activation functions and architectures.** We thank the reviewer for raising this important point. While our mechanistic construction in Sec. 2.2 focuses on ReLU networks for clarity, the underlying principle is more general. Encoding absences requires (i) inhibitory contributions from features representing a concept and (ii) a baseline level of activation that can be suppressed when the concept is present (referred to as "positive potential" in the ReLU case). This mechanism is not specific to ReLU and can be implemented by other activation functions. Please note that the two conditions can, in principle, also be inverted (L. 708–710).
>
>    In Appendix A.4, we discuss extensions to functions such as Tanh and leaky ReLU. As for Swish and GeLU activation functions, they can be seen as smooth variants of ReLU, and as such, they are capable of implementing the same mechanisms for encoded absences as outlined for ReLU activation functions.
>
>    Regarding architectures, we expect similar behavior in models beyond CNNs. To support this, we conducted additional experiments on Vision Transformers (ViTs), which confirm that encoded absences also arise in these models (see reply to Q3 of Reviewer RGDo). This provides empirical evidence that the phenomenon is not specific to convolutional architectures.
>
>    We will expand this discussion with more theoretical details and include these results in the revision to better reflect the architectural generality of our framework.

---

> > ### Author Rebuttal · Reviewer_6gdg · 2026-04-03
> >
> > Thank you for your detailed responses. Your rebuttals to **Reviewer mF5n** and **Reviewer RGDo** have addressed my concerns regarding **Q2** and **Q3**. However, upon reviewing the feedback from other reviewers and your subsequent replies, a few additional questions have emerged:
> >
> > You emphasized that "positive potential" remains stable in images; however, in your reply to **jAf9**, you admitted that the inhibitory effect in intermediate blocks is notably smaller. If these intermediate layers rely predominantly on presence-based reasoning, how is the high-level "absence encoding" transmitted across these layers that do not seem to encode absences themselves? Could you provide some evidence or analysis?
> >
> > If this question can be clarified, I would be willing to further increase my score.

---

> > > ### Author Response · Authors · 2026-04-05
> > >
> > > We again thank the reviewer for their valuable feedback and the follow-up.
> > >
> > > First, it is important to note that our results do not suggest that intermediate layers lack absence encoding, but rather that the inhibitory effect is *less pronounced* compared to early and late layers. In our analysis, the "+Least act." condition still consistently yields lower activations than the random baseline across all examined layers, indicating that absence-related signals are present throughout the network, albeit to varying degrees.
> > >
> > > Second, regarding the transmission of encoded absences: in many architectures – including the ResNet studied here, but also models like Vision Transformers – information is propagated not only sequentially but also via skip connections. Our layer-wise measurements from our reply to Q1 of Reviewer jAf9 are taken at the output of individual convolutional blocks *before residual addition*, whereas in the forward pass, these representations are combined with the skip pathway. This allows features, including encoded absences from earlier layers, to be preserved and integrated at later stages. We will make this point clearer in the revision.
> > >
> > > Independent of the above points, we note that strong absence encoding in later layers does not necessarily require equally strong absence encoding in all preceding layers. As shown by our mechanistic construction (Sec. 2.2), absences can even be implemented in layer $l$ when layer $l-1$ only encodes presences.

---

### Official Review · Reviewer_mF5n · 2026-03-13

**Soundness:** 3
**Presentation:** 4
**Significance:** 3
**Originality:** 3
**Overall Recommendation:** 4
**Confidence:** 4

**Summary:**

Overall, this study discusses a core problem in explainable AI: that existing explanation methods largely focus on presence-based reasoning while overlooking the role of absent concepts in neural network decision-making. The authors outline a broad topic by introducing the notion of encoded absences, where the absence of a concept causally increases neuron activation.

The paper makes three main contributions:

1.	A causal definition of encoded absences within a structural causal model framework.
2.	A mechanistic argument showing how standard neural networks can encode absence via inhibitory connections.
3.	Simple modifications to existing XAI tools: non-target attribution and feature visualization via minimization

Empirical validation is conducted via synthetic toy setups (including a Hassenstein–Reichardt detector-inspired model), ImageNet models showing prevalence of encoded absences, and a debiasing case study demonstrating practical utility.

The core claim is that standard XAI methods fail to capture absence-based reasoning, and incorporating it leads to more complete explanations and improved debiasing.

**Compliance With Llm Reviewing Policy:**

Affirmed.

**Final Justification:**

The first round of rebuttal has answered or clarified most of the questions, and there was no second round of rebuttal. Based on the overall evaluation of the paper's contribution and merits, and also on some of other reviewers' comments, I keep my overall assessment.

**Key Questions For Authors:**

1.	Can you provide large-scale evidence (e.g., across ImageNet) showing how frequently encoded absences materially affect predictions (not just neuron activations)?

2.	Causal validation:
- How robust is Definition 2.1 to stronger intervention tests beyond patch insertion?
- Could causal scrubbing or counterfactual generation validate the claim more rigorously?

3. How are “concepts” operationalized in real models? Can your framework integrate with automatic concept discovery methods?

4.	How does your approach compare quantitatively with TCAV-like or concept bottleneck methods in identifying absence-based reasoning?

5.	Can you demonstrate a practical pipeline that scales to large datasets without quadratic attribution cost?

**Limitations:**

Yes.

**Strengths And Weaknesses:**

### Strengths

1. Soundness

- The causal formulation (Definition 2.1) is clear and logically consistent.
- The mechanistic construction (negative weights + positive potential) is simple but convincing.
- Empirical results are consistent with claims: toy examples clearly isolate the phenomenon; ImageNet experiments show measurable activation suppression (Table 1).
- The authors noted limitations such as attribution noise and computational cost.

2. Originality

- The core conceptual contribution is novel and important: “encoded absence” is not explicitly formalized in prior XAI literature.
- The insight that standard methods fail not because of algorithmic limitations, but because of how they are used
is particularly strong.
- Reinterpreting non-target attribution and feature minimization, as absence-revealing tools is a clever reframing of existing methods.

3. Significance

The work addresses a fundamental blind spot in XAI.
Potential impact may include better interpretability (especially fine-grained classification), improved debiasing (demonstrated empirically), and conceptual implications for causal interpretability.

The idea may influence future interpretability research, especially neuron-level analysis, causal XAI frameworks, robustness and bias studies.

4. Presentation

The paper is well-written and logically structured. Figures (e.g., Fig. 3–6) are effective. The narrative flow is clear. Related work section is thorough and positions the contribution well.

### Weaknesses

1. Limited empirical depth (major)
- Experiments are mostly illustrative rather than rigorous:
Heavy reliance on toy setups,
ImageNet analysis is largely qualitative (Fig. 6)
- No large-scale quantitative study to show (1) how often encoded absences affect predictions, (2) impact on downstream performance (beyond one debiasing setup).
- The debiasing experiment is synthetic and narrow.

2. Incremental methodological contribution
- The proposed “methods” are non-target attribution (already used implicitly in adversarial settings) and feature minimization (previously explored)
- The novelty lies more in interpretation than algorithmic innovation.

So it appears more like a conceptual paper with minor technical novelty.

3. Causal claims are somewhat overstated
- The paper frames encoded absence as a causal relationship, but the SCM abstraction is not deeply validated, and empirical verification relies on attribution signals (which are not causal in general).
- No intervention-based validation beyond patch insertion (which may introduce artifacts).

4. Ambiguity in “concept” definition

The notion of a “concept” (x̂) is underspecified:
Is it human-defined?
Learned feature?
Direction in representation space?

5. Scalability concerns
- Non-target attribution scales poorly with number of classes, and dataset size
- The proposed mitigation is speculative and not evaluated.

6. Missing comparisons to stronger baselines
- Limited comparison with: concept-based methods (e.g., TCAV-like approaches), recent neuron interpretability / circuit analysis work
- The “failure” of existing methods is demonstrated, but not exhaustively.

---

> ### Author Rebuttal · Authors · 2026-03-30
>
> We thank the reviewer for the constructive feedback and answer the key questions raised below:
> - **Q1: Impact on predictions in large-scale settings.** To test whether encoded absences affect predictions (not only activations), we ran a large-scale ImageNet experiment on all correctly classified validation images of ResNet-50. For each image, we identify the most important penultimate-layer channel (via InputXGradient), retrieve its 8 least-activating patches, and paste them into border locations. As a control, we also insert 8 random patches at the same locations.
>
>    This yields a flip rate of 84% vs. 8% (least-activating vs. random). Since least-activating patches can introduce presence-based evidence, we also consider a suppression-only setting, which ensures that no neuron's activation increases relative to the original image. Under this constraint, the flip rate remains at 8.2% vs. 3.5% (least-activating vs. random), indicating that absence-related (inhibitory) signals still materially contribute to predictions at ImageNet scale.
>
>    We note that we use only the single most important channel for finding the 8 least-activating patches, making this a conservative estimate; using multiple channels would likely strengthen the effect. We will add this experiment to the revision.
> - **Q2: Causal validation beyond patch insertion.** Definition 2.1 is formulated in terms of an idealized intervention (see reply to Q2 of Reviewer 6gdg), and we agree that stronger intervention schemes – such as causal scrubbing or counterfactual generation – would provide a more direct test of this definition. To this end, in Appendix B.5 (Fig. 12), we go beyond simple patch insertion and consider stronger interventions that better preserve context. Across these settings, we observe consistent inhibitory (negative) attribution patterns, providing consistent evidence beyond patch insertion. However, such interventions are difficult to implement reliably in natural images. Thus, we focus in the main paper on simple, controlled interventions (patch insertion) combined with random-patch baselines, which allow us to separate inhibitory effects from content overlays.
>
>     Overall, we view our evidence as consistent with the interventional notion in Definition 2.1 across multiple intervention types, while acknowledging that stronger interventions remain an important direction for future work that will be discussed in the revision.
> - **Q3 & Q4: Concept discovery methods & quantitative comparison to them.** Consistent with prior XAI work, in our framework, "concepts" are operationalized as *input patterns* that influence neuron activations. While Definition 2.1 is formulated at the level of specific inputs/interventions, we observe that the identified patterns generalize across semantically related inputs, consistent with common XAI assumptions. As noted in our reply to RGDo (Q1), many concept discovery methods primarily capture the presence of concepts and therefore may not directly reveal encoded absences when applied in their standard form. The same applies to TCAV, where a concept activation vector (CAV) corresponds to the presence of a concept. This is reflected, e.g., in Sec. 4.1.2 of the TCAV paper (Kim, 2018), where image patterns are optimized to maximally activate the CAV. As discussed for feature visualization (L. 165–176), this primarily reveals encoded presences.
>
>    Our framing is thus complementary rather than competing. To capture encoded absences, these methods would require extensions (e.g., minimally activating examples or explicit inhibitory probes), analogous to our adaptations in Sec. 4. For this reason, a direct quantitative comparison in their current form is not well aligned with the notion of absence-based reasoning that we study. Instead, we view our work as a conceptual extension that can be integrated into concept-based methods, and we will clarify this connection more explicitly in the revision.
>
> - **Q5: Scaling to large datasets without quadratic attribution cost.** To address this point, we run an additional experiment, following the two-stage procedure briefly outlined in L. 425–428: Specifically, for each penultimate-layer channel of a ResNet-50, we select the 10 least-activating patches from ImageNet (and their source images), and compute non-target Integrated Gradients (IG) only on this small subset, using 10 random images as a control. This avoids exhaustive attribution over the full dataset. If the selected images contain encoded absences, we expect stronger negative attributions than for random images. Indeed, the results show that the average negative attribution is substantially higher (178.25 vs. 112.44), showing that meaningful non-target attributions can be recovered without quadratic cost. We note that the average negative attribution of random images is still substantial, as they may contain similar suppressive features and IG is noisy. This experiment will be included in the revision.

---

> > ### Author Rebuttal · Reviewer_mF5n · 2026-04-03
> >
> > Q1 (importance at prediction level) is improved by the new ImageNet experiment showing the 84% vs. 8% flip rate (least-activating vs. random). The suppression-only control (8.2% vs. 3.5%) directly targets the concern about disentangling absence vs. presence effects. Q5 (scalability) is reasonably addressed as the proposed two-stage pipeline (minimization → selective attribution) is now empirically supported, while not fully mature.
> >
> > However, Q2/3/4 remain partially resolved:
> >
> > Q2 (causal validation): The evidence still relies largely on attribution signals, which are not causal. The “stronger interventions” are mentioned but not clearly quantified or compared. Would you provide a more concrete comparison across intervention types (e.g., patch vs. stronger interventions) to support the causal interpretation?
> >
> > Q3-4: The response clarifies that concepts is “input patterns influencing neurons,” but this remains underspecified for modern XAI standards. The argument that comparison to TCAV-like methods is “not well aligned” is not fully convincing: even if imperfect, empirical comparison would strengthen the paper; the claim that such methods cannot capture absences is asserted rather than demonstrated. So, is it possible for you to provide a quantitative comparison (even partial) showing that TCAV or similar methods fail to capture absence effects in practice?

---

> > > ### Author Response · Authors · 2026-04-05
> > >
> > > We again thank the reviewer for their valuable feedback and the follow-up.
> > >
> > > **Q2.** We agree that comparing intervention types strengthens the causal interpretation, and thus evaluate whether they produce consistent inhibitory effects. In particular, we consider:
> > > - (i) patch-based interventions (as in Sec. 5.3), and
> > > - (ii) controlled interventions following the data-generating process (for the toy and ISIC setups in Sec. 5.2 and 5.4), which approximate the counterfactual generation proposed by the reviewer.
> > >
> > > For both setups, we consider the output neuron known to encode an absence, take its corresponding class images, and include the hypothesized concept whose absence is encoded by the neuron. We further report the corresponding negative attributions to verify that attribution signals behave consistently under these interventions.
> > >
> > > Results:
> > >
> > > ||Original sample without intervention||w/ controlled intervention||w/ patch intervention||
> > > |---|---:|---:|---:|---:|---:|---:|
> > > |**Dataset**|**Avg. activation**|**Avg. neg. attr.**|**Avg. activation**|**Avg. neg. attr.**|**Avg. activation**|**Avg. neg. attr.**|
> > > |Toy|0.22|0|-0.11|10.44|-0.11|10.43|
> > > |ISIC|-0.06|926.55|-0.73|1469.09|-0.98|1501.5|
> > >
> > >
> > > Across both settings, we observe that controlled and patch-based interventions lead to similar and *consistent suppression effects*, and the magnitude of negative attributions increases accordingly, as expected.
> > >
> > > For ImageNet-1k, controlled interventions are not available due to the lack of access to the data-generating process; however, consistent evidence from patch-based interventions (Sec. 5.3) and attribution-based analysis (our reply to Q5) supports the same conclusion.
> > >
> > > Overall, while all interventions remain approximations of the ideal do-intervention (cf. Appendix A.2), their consistent behavior across different settings provides empirical support for the causal interpretation in Definition 2.1. We will include this analysis in the rebuttal.
> > >
> > > **Q3.** In our framework, and consistent with existing concept discovery methods such as TCAV or WWW (Ahn, 2024; see RGDo) – which, despite their differences, ultimately ground or validate concepts through sets of images – we define a concept as a set of input patterns, which we typically assume to be semantically related (e.g., images of "dog snouts") for interpretability purposes. Strictly speaking, however, a concept in our formal definition (Definition 2.1) can be any family of input patterns that share a common effect on a neuron – semantic coherence is a natural and practical assumption but not a formal requirement. In Appendix A.4, we further discuss how the framework extends to arbitrary feature space directions, which typically correspond to more semantically coherent concepts than individual neurons. Semantic interpretations (e.g., the *word* "dog snout") are post-hoc descriptions of such families, rather than part of our formal definition itself.
> > >
> > > In our experiments, concepts are operationalized via feature visualization through minimization or image regions identified through negative attributions; in principle, more sophisticated approaches, such as TCAV or WWW, could also be used to identify such pattern families. However, as discussed in our previous reply and empirically demonstrated in the point below, such methods would require appropriate adjustments to be compatible with encoded absences.
> > >
> > > We will revise the paper to make these points more explicit.
> > >
> > > **Q4.** To address this point, we evaluate WWW, which operates at the neuron level and does not require predefined concept annotations, making it comparable to our setting. Briefly, WWW identifies concepts by grouping highly activating inputs (e.g., image patches) for a neuron and assigning semantic labels (cf. their Fig. 1 & 3). Since our focus is on the underlying input patterns, we operate directly on the retrieved image sets and omit the labeling step.
> > >
> > > To remain consistent with our setup, we use 48x48 patches from ImageNet validation images as the probing dataset. Following Sec. 5.3/Tab. 1, we insert the identified patterns into the highest-activating images for each neuron and measure the resulting change in activation, yielding the following results:
> > >
> > > | Method | VGG19 | ResNet-50 |
> > > |---|---:|---:|
> > > |None | 2.98| 0.18 |
> > > |+ Random | 2.68 | 0.16 |
> > > |+ Least activating (ours) | 0.94|0.03 |
> > > |+ WWW | 3.88 | 0.25|
> > >
> > > Supporting our earlier argument, patterns identified by WWW increase activation, consistent with its focus on presence-based evidence. In contrast, our method identifies patterns whose insertion suppresses activation, corresponding to encoded absences. This empirically supports our claim that standard concept extraction pipelines, when applied in their proposed form, primarily recover presence-based encodings and are not designed to capture absence-based encodings. We will include this analysis in the revision.

---

### Decision · Program_Chairs · 2026-04-30

**Decision:**

Accept (regular)

**Comment:**

This XAI / interpretability article focuses on a type of features that aren't typically studied: encoded absences, i.e., cases where the absence of a concept leads to an increase in activation. This is a novel and refreshing perspective that will be of interest to the community. During the rebuttal discussion, authors were able to address and clarify important concerns and added new results (e.g. on ViTs); as a result, all reviewers are unanimously in favor of acceptance.